# LncRNAs and microRNAs as Essential Regulators of Stemness in Breast Cancer Stem Cells

**DOI:** 10.3390/biom11030380

**Published:** 2021-03-03

**Authors:** Nadia Flores-Huerta, Macrina B. Silva-Cázares, Lourdes A. Arriaga-Pizano, Jessica L. Prieto-Chávez, César López-Camarillo

**Affiliations:** 1Laboratorio de Oncogenómica y Proteómica del Cáncer, Posgrado en Ciencias Genómicas, Universidad Autónoma de la Ciudad de México, 03100 CDMX, Mexico; fnadia3121@gmail.com; 2Doctorado Institucional en Ingeniería y Ciencias de los Materiales, Universidad Autónoma de San Luis Potosí, 78210 San Luis Potosí, Mexico; macrina.silva@uaslp.mx; 3Unidad de Investigación Médica en Inmunoquímica, Hospital de Especialidades del Centro Médico Siglo XXI, Instituto Mexicano del Seguro Social, 06720 CDMX, Mexico; landapi@hotmail.com; 4Laboratorio de Citometría de Flujo, Centro de Instrumentos, Coordinación de Investigación en Salud, Hospital de Especialidades del Centro Médico Siglo XXI, Instituto Mexicano del Seguro Social, 06720 CDMX, Mexico; lakshmi.litmus@hotmail.com

**Keywords:** breast cancer, cancer stem cells, microRNAs, lncRNAs, therapy

## Abstract

Breast cancer is an aggressive disease with a high incidence in women worldwide. Two decades ago, a controversial hypothesis was proposed that cancer arises from a subpopulation of “tumor initiating cells” or “cancer stem cells-like” (CSC). Today, CSC are defined as small subset of somatic cancer cells within a tumor with self-renewal properties driven by the aberrant expression of genes involved in the maintenance of a stemness-like phenotype. The understanding of the underlying cellular and molecular mechanisms involved in the maintenance of CSC subpopulation are fundamental in the development and persistence of breast cancer. Nowadays, the hypothesis suggests that genetic and epigenetic alterations give rise to breast cancer stem cells (bCSC), which are responsible for self-renewal, tumor growth, chemoresistance, poor prognosis and low survival in patients. However, the prominence of bCSC, as well as the molecular mechanisms that regulates and promotes the malignant phenotypes, are still poorly understood. The role of non-coding RNAs (ncRNAs), such as long non-coding RNAs (lncRNAs) and microRNAs (miRNAs) acting as oncogenes or tumor suppressor genes has been recently highlighted by a plethora of studies in breast cancer. These ncRNAs positively or negatively impact on different signaling pathways that govern the cancer hallmarks associated with bCSC, making them attractive targets for therapy. In this review, we present a current summary of the studies on the pivotal roles of lncRNAs and microRNAs in the regulation of genes associated to stemness of bCSC.

## 1. Introduction

Breast cancer (BC) is an aggressive illness that compromises the physiology and normal functions of mammary glands due to the uncontrolled proliferation and growth of transformed epithelial cells. This neoplasia is the second most common type of cancer worldwide. In 2018, it was estimated that its incidence was approximately 2 million diagnosed cases around the world, resulting in 626,679 cancer related-deaths [1,2]. Precise and opportune diagnosis is necessary to allow a favorable prognosis and increased survival rates [3].

Long ago, it was hypothesized that cancer arises from genetic alterations that produce subpopulations of “cancer stem cells-like” (CSCs) or “tumor initiating cells” which are considered to be essential protagonists of tumor initiation and progression, chemoresistance, and cancer recurrence after chemotherapy. This hypothesis has been difficult to confirm, and an alternative hypothesis postulates that CSCs originate from genetic and epigenetic alterations in mature differentiated cells, which drives the activation of a set of genes involved in pluripotency, stemness, and development during the early stages of tumorigenesis. Currently, there is also controversy regarding the exact nomenclature for cells with stemness properties, as most research denominates these subpopulations as either “cancer stem cells-like” or “tumor initiating cells” (here, in this review, we refer to the two subpopulation types as synonymous). Thus, the origin of CSCs is uncertain; however, three possible hypotheses are viable: (i) through the malignant transformation of normal stem cells; (ii) de-differentiation of cancer cells through the epithelial–mesenchymal transition (EMT) or; (iii) through the induction of pluripotent cancer cells [4]. In the same way that stem cells exist in normal tissues, CSCs are also present in tumor tissues where they possibly originated through genetic and epigenetic mutations that promoted specific failures in cellular processes [5]. Independently of their origin, CSCs are defined as cells with the ability to preserve themselves as result of self-renewal and with the capability of generating mature cells of a determined tissue through cell differentiation [5,6]. 

In recent years, knowledge of CSCs as an essential factor in the development of BC has emerged. It has been demonstrated that breast CSCs (bCSCs), or breast tumor initiating cells, contain high levels of CD44 and low levels of CD24 proteins. Although bCSCs with an CD44^+^/CD24^−/low^ immunophenotype represent a minor subpopulation in bulk tumors, these cells have the ability to produce solid tumors efficiently in mice when injected and to propagate in vitro through extensive cell proliferation and self-renewal [7,8]. Knowledge of the key role of bCSCs in tumor progression and chemoresistance has led to interest in the development of novel strategies and therapies targeting specific molecular mechanisms that regulate the signaling pathways involved in malignant transformation in order to stop their dissemination. However, despite being overexpressed in CSCs, many proteins involved in stemness maintenance are also found in normal cells or have low immunogenicity, which explains, in part, the limited therapeutic interventions [9,10]. In this regard, non-coding RNAs, including long non-coding RNAs (lncRNAs) and microRNAs (miRNAs), have been studied in depth as promising tools for bCSC therapy, as they are the main regulators of genes operating in CSC phenotypes. 

MiRNAs are small, non-coding, single-stranded RNA molecules that modulate gene expression at the post-transcriptional level through recognition and binding to specific 8-mer sequences in the 3′-UTR region of target messenger RNA (mRNA) [11,12]. On the other hand, lncRNAs are non-coding, single-stranded molecules that are involved in the regulation of multiple molecular processes, mainly through epigenetic, transcriptional, and post-transcriptional modulation of genes and proteins; both play pivotal roles in development, pluripotency, growth, angiogenesis, apoptosis, chemoresistance, self-renewal, and metastasis in CSC subpopulations. This review describes the origin, cell markers, and signaling pathways operating in bCSCs and then evaluates the use of strategies to eliminate bCSCs using miRNAs and lncRNAs as therapeutic tools. Finally, a limited number of clinical trials using miRNAs-based approaches are depicted.

## 2. The Mammary Glands: Main Biological and Molecular Characteristics

The mammary glands are the tissues that distinguish mammals from other groups of animals. Their main function is to produce and secrete milk to support a species’ descendants. The glands undergo many changes throughout a woman’s life, which are induced by hormonal changes during the menstrual cycle, pregnancy, and lactation [13]. The epithelium of the mammary glands is composed of apically-oriented luminal cells and basally-oriented, elongated myoepithelial cells that are in contact with the basement membrane. It has been proposed that, in this structure, there are mammary stem cells (MaSCs) with both luminal and basal lineages. Several cell surface markers have been used for the identification of MaSCs in human and mice mammary tissues. This includes epithelial cell adhesion molecule (EpCAM), Procr, sca-1, Myh11, CD61, CD133, CD10, CD24, CD29, CD49f, CD49b, c-Kit, THY1, Lrp5, Axin2, CK5, CK8, CK14, CK18, CK19, Lgr5, Lgr6, and CD1d [14]. The mammary gland tissue has a high regeneration capacity, high proliferative capability, and plasticity due to its capacity to generate the whole organization’s epithelial and cell complex hierarchy, whose organization and functioning depends on diverse signaling pathways that maintain cellular homeostasis [15]. The Wnt/β-catenin, Notch, Hippo, and Hedgehog pathways are actively involved in the development of the breasts, in terms of their morphogenesis and functioning [15]. The Wnt pathway is involved in the growth, development, maintenance, and morphogenesis associated with the normal physiology of MaSCs as well as with tumor development [16,17,18,19]. Notch signaling participates in the regulation of self-renewal, differentiation, or cell fate, depending on the cellular and development context. The Notch pathway acts as a regulator of cell survival and cell proliferation and even may act as a proto-oncogene [20,21,22]. The Hippo-YAP pathway is an evolutionarily conserved pathway that exerts an important role in the control of organ size through the regulation of proliferation, apoptosis, and self-renewal; moreover, it is involved in the mammary tumorigenesis induced by epidermal growth factor receptor (EGFR) [23,24,25]. The roles of several signaling pathways in bCSCs are discussed later in Section 4.

In BC, tumor progression is associated with the disruption of this epithelial organization and polarity [26]. This organization is related to the asymmetric division that produces two daughter cells with different cellular fates: a copy of the original SCs and another programmed to differentiate in a dissimilar type of cell [27]. The *p53* gene is a tumor suppressor that regulates cellular division and is also called the “guardian of the genome” as it regulates the polarity of cell division in, and continuous expansion of, MaSCs. Furthermore, its loss favors symmetric divisions of CSCs, contributing to tumor growth [28]. It is important to mention that the loss of *p53* in both lineages, luminal and basal MaSC progenitors, results in an unlimited and increased capacity to proliferate and undergo self-renewal [29,30]. Other important transcription factors, known as master regulators of pluripotency, e.g., c-Myc, Nanog, Sox2, and Oct4, proteins are known to be regulators of the embryonic stem cell state and have been identified as being overexpressed in multiple tumors. The upregulation of embryonic transcriptional factors may be associated with less differentiated tumors; thus, they are responsible for maintaining the undifferentiated state, and the production of these cells requires overexpression of the aforementioned transcription factors. For example, bCSCs and luminal progenitors express high levels of the transcription factor Myc, which exerts an indispensable function in the maintenance of self-renewal [31,32,33,34,35]. Therefore, based on this information, it is essential to elucidate the molecular mechanisms of normal MaSCs and to understand the alterations in signaling pathways that may contribute to mammary carcinogenesis in transformed cancer cells, favoring the appearance of bCSCs.

## 3. Breast Cancer Stem Cells Origin and Their Main Cell Markers

BC is characterized by a high degree of intratumoral heterogeneity, so its classification varies between patients, which exerts a direct effect on the selection of treatment, so this heterogeneity determines tumor evolution [14]. The precise origin of bCSCs is ambiguous and has been controversially debated for a long time. Several similar features exist between cancer cells and bCSCs. Both types of cell can self-renew and share signaling pathways associated with cell replication and maintenance [14]. Particularly, two nonexclusive BC models have been proposed to explain the presence of clonal populations in tumors. The first model involves the stochastic appearance of mutations and clonal selection that grant the cells stem-like properties and the ability to differentiate and self-renew. In the second model, the MaSCs and progenitor attributes are central to the heterogeneity of the BC cell populations. The accumulation of genetic and epigenetic alterations results in the development of at least one cell with CSC traits that can produce more CSCs and more differentiated offspring. In the past, the CSC model was thought to be a static one; in recent times, it has been revised to a dynamic one, where CSCs are believed to convert into more transient cell types [14,36]. It is important to mention the heterogeneity of cancer cells; not all cancer cells are stem cells or exhibit properties similar to stem cells. Cell diversity and heterogeneity is a product of the mutagenesis present in cancer cells and results in incomplete or aberrant hierarchical cellular differentiation. Therefore, according to the clonal evolution/stochasticity, all tumoral cells have a similar tumorigenic potential and tumor heterogeneity arises as a result of the generation of intra-tumoral clones through sequential mutations. This model presumes that bCSCs can be generated from differentiated mammary cells by virtue of mutations that occur during the course of the disease. Exposure to detrimental environmental factors, such as radiation and chemotherapies, induces genetic alterations in non-malignant somatic cells that prime the de novo generation of bCSCs through the de-differentiation process, and microenvironmental signals can even induce the malignant transformation of differentiated cells into bCSCs. The hierarchical or CSC model postulates that only a small proportion of tumor cells residing in the tumor have tumor-propagating potential. These cells exhibit self-renewal properties and are capable of reiterating tumor hierarchy [14,36,37]. The bCSCs are on top of a hierarchy of cells that form the tumor. At a determinate moment when the microenvironment is ideal, they proliferate, allowing them to evade chemotherapy and stimulating the recurrence of tumors [15,38]. In addition, the current gold standard method to identify CSCs is via serial transplantation of cancer cells in animal models. In this regard, bCSCs with an CD44^+^/CD24^−/low^ immunophenotype produce solid tumors efficiently in mice, and the CD44^+^/CD24^−^ signature is a molecular determinant of bCSC [7,8].

Studies have suggested that there is a close association between CSCs and the acquisition of an EMT state [39]. Interestingly, bCSCs exist in distinct mesenchymal-like EMT and mesenchymal-epithelial transition (epithelial-like, MET) states. Mesenchymal-like bCSCs, characterized as CD24^−^/CD44^+^, are primarily quiescent and are localized at the tumor invasive front, whereas epithelial-like bCSCs express aldehyde dehydrogenase (ALDH) proliferatively and are located more centrally [40]. However, further studies are needed to more fully define the relationship among EMT, MET, and bCSCs. Furthermore, when the oncogene Twist, a transcription factor and the main regulator of the mesenchymal phenotype, is overexpressed, it promotes the transformation of MCF-7 breast cancer cells into fibroblasts that have morphology characteristic of a mesenchymal phenotype, which is pivotal in the transformation to aggressive BC [41]. Similarly, the transient expression of Twist in several breast cancer cell lines induces transformation to a stem cell phenotype and transcriptionally regulates CD24 expression in bCSCs [42]. Essentially, the EMT program assists in cell survival and immune tolerance and promotes chemoresistance in bCSCs [43]. Moreover, it is known that the EMT program is fundamental for the activation of tumoral invasion and metastasis processes in BC. Studies have demonstrated that the generated SCs acquire characteristics typical of normal SCs, such as self-renewal, fibroblast-like, mammosphere formation, and mesenchymal appearance, and at the transcriptional level, the up-regulation of mRNA encoding mesenchymal markers. In conclusion, the induction of the EMT in normal and tumor mammary epithelial cells promotes the formation of SCs with the mesenchymal-like phenotype [39,44].

The accepted markers for bCSC recognition and isolation are CD44^+^ and CD24^−/low^ proteins, which are expressed in the epithelial cell surface. However, their expression is not always consistent, even within the same BC sub-type [45,46]. CD44 is a glycoprotein that acts as an adhesion molecule. It is mainly found in primary breast carcinomas and is aberrantly expressed, depending on the level of cell differentiation. BCs with a basal phenotype express a high percentage of bCSCs with the CD44^+^/CD24^−/low^ immunophenotype, because they probably originate from the most primitive mammary stem cells, contrary to the luminal phenotype that shows cell population enrichment with CD44^−^/CD24^+^ [47,48,49]. Interestingly, the cell lines that present a great percentage of CD44^+^/CD24^−^ also express high levels of genes associated with invasion [50]. CD24 is a small, heavily glycosylated mucin-like glycosyl phosphatidyl inositol-linked cell surface protein that is expressed in a wide variety of human malignancies, including BC. This protein is considered a prognosis marker, as its expression might enhance the metastatic potential of tumor cells [45]. In recent years, other markers have been identified, including aldehyde dehydrogenase 1 (ALDH1), cluster of differentiation 133 (CD133), cluster of differentiation 49f (CD49f), and cluster of differentiation 90 (CD90), which is a glycosyl phosphatidyl inositol (GPI)-anchored glycoprotein. The presence of these proteins is often associated with chemotherapy and radiotherapy resistance. In addition, combinations of these markers have been used to improve the prognostic value (Table 1) [36,38].

The function of CD44^+^/CD24^−/low^ cells in BC was demonstrated in samples of nine patients, where they were showed to induce tumor development in NOD/SCID mice, in contrast to the use of cells with a different immunophenotype [7]. These observations were supported by the in vitro propagation of bCSCs isolated from breast carcinoma cell lines and breast tumor lesions. The authors reported that these cells conserve their self-renewal and great proliferative capability and have the ability to form new tumors in mice [8]. Moreover, these findings were confirmed using Oct-4, which is a putative SC marker. Oct-4 is a master regulator that is fundamental in self-renewal and pluripotency and is associated with the maintenance and expansion of CSCs [8,59,60]. In this regard, several studies have been performed with the purpose of identifying diverse antigens or markers present in bCSCs. On the other hand, the ALDH1 enzyme has been identified as a potential marker for human bCSC. High expression of this enzyme is associated with early metastasis, poor prognosis, and low survival of patients [51,61]. Another positive marker found in bCSCs is glycosylated trans-membrane protein CD133/prominin 1, which is involved in membrane organization [52]. It is considered an important indicator of malignancy, as both high and nuclear expression may be associated with a poor prognosis and outcome for BC patients. However, even though high expression of CD133 in triple negative MDA-MB-231 breast cancer cells is related to a great invasion capability, studies suggest that CD133 can be used as another target for immunotherapy [52,53,54,62]. The cluster of differentiation 49f (CD49f) is an α6-integrin that promotes epithelial cell adhesion to ECM, facilitating signal transduction pathways and the communication cell-ECM. Its expression is associated with cancer recurrence, poor prognosis, and a low survival rate in breast cancer patients [38,55,56]. Recently, CD90, a GPI-anchored glycoprotein that interacts with integrins, was identified as stem cell marker. It is induced by the EMT and possibly also by immune cells in the bCSC tumor microenvironment [38,57,58].

## 4. Signaling Pathways Governing the Breast Cancer Stem Cell-Like Phenotype

As previously mentioned, there are various signaling pathways involved in the correct functioning and regulation of the biological processes of bCSCs. The main cellular pathways include NF-κB, PI3K/AKT/mTOR, Notch, Hedgehog, Wnt, Hippo, JAK/STAT, TGF-β, SMAD, PPAR, and signaling mediated by the estrogen receptor (ER). The deregulation of these pathways is one the main reasons for the appearance or exacerbation of the main hallmarks of bCSCs, such as maintenance, self-renewal, tumor resistance, recurrence, and metastasis. Additionally, the tumoral microenvironment factors, such as the vascular niche, MEC, hypoxia, and immune cells, play important roles in the regulation of bCSCs [36,38,63].

### 4.1. The Notch Pathway

The Notch pathway is highly conserved and participates in the proliferation and maintenance of SCs. Notch is recognized as functioning as both an oncogene and a tumor suppressor gene in a form that depends on the microenvironment. Studies on the Notch pathway in CSCs have shown that the activation of Notch promotes cell survival, self-renewal, and metastasis. Aberrant overexpression of Notch4 signaling by 8-fold and Notch1 by 4-folds was observed in bCSCs. Interestingly, Notch inhibition produces a decrease in the number and size of tumors [64]. In another study, it was found that Notch4 contributes to the maintenance of mesenchymal-like bCSCs. The transcriptional regulation of SLUG and GAS1 by Notch4 promotes quiescence and the EMT in breast cancer cells. This study contributes to the knowledge about how Notch4 regulates stemness, chemoresistance, and the invasion of bCSCs [65].

This pathway consists of the Notch receptor (Notch1–4), Notch ligand (Delta-like 1, 3, and 4, Jagged 1 and 2), CSL (CBF-1, suppressor of hairless, Lag), Notch regulatory molecules, DNA-binding proteins, and other effectors. Notch receptors and DSL ligands are transmembrane proteins that mediate communication between adjacent cells. The ligand binds to a Notch receptor that is expressed on adjacent cells, thus triggering proteolytic cleavage of the Notch intracellular domain (ICD). Translocation into the nucleus occurs to bind to the transcription factor CSL, forming the NICD/CSL transcriptional activation complex, which activates target genes of the bHLH transcription inhibitor family [63,66,67,68]. In general, oxygen homeostasis in BC is an important protagonist of tumor evolution and carcinogenesis. Hypoxia plays a critical role in the promotion of the aggressive development of the tumoral structure, contributing to angiogenesis and metastasis. This evidence supports the idea that Jagged 2 is upregulated by hypoxia and promotes Notch activation, which, in turn, triggers the EMT program. Consequently, Jagged2 overexpression is related to poor prognosis and metastasis [69].

### 4.2. The Hedgehog Pathway

The Hedgehog pathway components PTCH1, Gli1, and Gli2 are highly expressed in normal human mammary stem/progenitor cells and are activated in human bCSCs. Previous investigations support the idea that the hedgehog pathway and Bmi-1 play important roles in regulating the self-renewal of normal and tumorigenic human breast SCs. This emphasizes the importance of the Hedgehog signaling pathway and Bmi-1 in the regulation of normal and malignant stem cells and suggests that strategies aimed at inhibiting these pathways represent a rationale therapeutic approach [70]. Tumoral heterogeneity is an important characteristic that allows tumor growth, recurrence, and therapy resistance. Cancer-associated fibroblasts (CAFs) support the development of CSCs. The participation of Hedgehog pathways has been demonstrated. The CSCs can regulate the CAF response through secretion of the Hedgehog ligand. As a consequence, the CAFs produce factors that promote the expansion and self-renewal of CSCs. Thus, inhibition of this signaling pathway could be used as a novel therapeutic strategy against BC [71].

This complex signaling pathway comprises transmembrane receptors (PTCH 1-2), extracellular Hh ligands, the transmembrane SMO protein, transduction molecules, and downstream molecules (GLI 1-3). These molecules play different roles, even between subtypes. For example, SMO plays a positive regulatory role, while the transmembrane protein PTCH plays a negative regulatory role. The Hh ligands bind to PTCH receptors, initiating a series of intracellular signal transduction reactions. When there is no ligand signal, the transmembrane receptor PTCH binds to SMO, inhibiting its activity. The Hh ligand is bound to PTCH, promoting changes in the spatial conformation of PTCH, and the inhibition of SMO is removed, thus activating the transcription factor to enter the cell nucleus, promoting cell growth, proliferation, and differentiation [63,66,67,68].

### 4.3. The Wnt Pathway

The Wnt system is a complex and evolutionarily conserved pathway that is fundamental to the regulation of proliferation, the determination of cell fate, and the integrity of the SC niche. Wnt signaling has been implicated in the regulation of bCSCs and is aberrantly expressed in BC [72]. In fact, it was demonstrated that the Wnt–β-catenin pathway induces Lin28 up-regulation and let-7 down-regulation, impacting the development of BC. This connection between the Wnt–β-catenin pathway and Lin28/let-7 is essential for bCSC expansion [73]. Further, focal adhesion kinase (FAK) and the actin-binding protein Fascin play relevant roles in activation of the β-catenin pathway. High expression of the Fascin–FAK–β-catenin axis is associated with poor survival outcomes, so it has become an important target for targeted therapy of bCSCs [74]. Nestin, an intermediate filament protein, has been associated with a malignant phenotype and poor prognosis in patients. Nestin is overexpressed in BC tissue; however, its participation in the development of bCSC is unknown. Nevertheless, it is associated with poor survival in patients with triple-negative BC. Its silencing inhibits bCSC invasiveness, cell cycle arrest, and apoptosis. It has been suggested that Nestin can stimulate the proliferation, survival, and migration of bCSCs by enhancing Wnt/β-catenin signaling [75]. The overexpression of CXCL12 promotes the proliferation, migration, and invasion of MCF-7. This overexpression is complemented by the up-regulation of vimentin, N-cadherin, α-SMA and Oct-4, Nanog, and Sox2. These results suggest that activation of the Wnt/β-catenin pathway induces CSC-like phenotypes through EMT programming in MCF-7 cells [76].

This pathway includes 19 Wnt ligands and more than 15 receptors. It is divided into the canonical Wnt pathway (FZD-LRP5/6 receptor complex, leading to derepression of β-catenin) and the non-canonical Wnt pathway. In the absence of Wnt ligands (inactive state), β-catenin is phosphorylated by glycogen synthase kinase 3β, which leads to β-catenin degradation, thereby inhibiting translocation of β-catenin to the nucleus. Dissimilarily, in the presence of Wnt ligands, the ligands combine with FZD receptors and LRP co-receptors (active state). LRP receptors are phosphorylated and β-catenin is released from the Axin complex to translocate to the nucleus. In the non-canonical Wnt signaling (FZD receptors and/or ROR1/ROR2/RYK co-receptors, activating PCP, RTK, or Ca^2+^ signaling cascades), β-catenin is not involved in Wnt/PCP signaling, and Dvl is activated through the binding of the Wnt ligands-ROR-Frizzled receptor and inhibits the binding of GTPase Rho-DAAM1. The GTPases Rac1 and Rho trigger ROCK (Rho-kinase) and JNK (c-Jun N-terminal kinase), which promotes transcriptional responses, cytoskeletal rearrangement, and cell growth [65,66,67,68].

In this regard, the Wnt pathway represents an attractive target for a rational design for the treatment of BC. Resistance to therapy is one of the main problems in the failure of BC treatment. Other strategies for the elimination of BC have been employed and analyzed. Clinical studies support the use of anti-parasitic treatment for BC elimination. For example, the anthelmintic pyrvinium pamoate (PP) inhibits and delays the proliferation of BC cells in vitro and in vivo, acting as a potent Wnt pathway inhibitor. In addition, self-renewal capability and EMT activation are suppressed, and conjointly, the populations of bCSC CD44^+^/CD24^−^ and ALDH^+^; Nanog, Sox2, and Oct4 diminish significantly following the action of PP. This evidence supports the idea that PP serves as a promising agent that targets bCSCs. Strategies combining PP with standard chemotherapy drugs can be used to efficiently eliminate bCSCs [77]. The combination of docetaxel (DTX) and sulforaphane (SFN)-loaded poly(D, L-lactide-coglycolide)/hyaluronic acid (PLGA-b-HA)-based nanoparticles used to simultaneously target differentiated BC cells and bCSCs enhances cytotoxicity. They are more effective by down-regulating β-catenin expression in bCSCs [78]. This attractive strategy is very promising and needs to be explored through the employment of several drugs.

### 4.4. The Hippo Signaling Pathway

The transcriptional co-activator TAZ is also necessary for the maintenance of the self-renewal capability, tumor initiation, and tumoral formation in bCSCs. High expression and activity of TAZ is linked with poorly differentiated tumors and to the generation of EMT programs in bCSCs [79]. In addition, it is known that TAZ is required for in vitro chemoresistance, which favors metastasis through cell migration. These facts corroborate the importance of TAZ as a mediator that is central to the metastatic capability of bCSCs, and it is suggested that TAZ could act as a potential biomarker and target for rational drug design [80]. In addition, YAP is a promising therapeutic target for bCSCs. Leukemia inhibitory factor receptor (LIFR) is capable of inhibiting metastasis activation through Hippo signaling by increasing YAP phosphorylation, which is conducive to its cytoplasmic arrest, evading its translocation to the nucleus and the activation of its transcription [81]. Prominent expression of YAP is crucial for stemness-related gene expression and is linked with invasiveness and stem-like signatures in BC patients. In this regard, the defeat of YAP suppresses oncogene-induced mammary tumors. This highlights the potential of YAP inhibitors to act as molecular therapies directed against BC [82]. In addition, pharmacological inhibition of YAP is an attractive strategy for the blockage of Hippo signaling. In fact, the antipsychotic drug Chlorpromazine suppresses stemness properties, such as mammosphere formation, the expression of ALDH, and stemness-related gene expressions in bCSCs. These effects are related to the suppression of YAP through the modulation of Hippo signaling and the promotion of YAP proteasomal degradation [83]. TAZ, another transcriptional activator, is a critical promoter of bCSC properties due to its in vitro/in vivo tumorigenic potential, which suggests that the pharmacological inhibition of TAZ activity may be a novel way to eliminate bCSC [84].

The core components of the canonical Hippo pathway are two kinase complexes: the first is the sterile 20-like kinase (MST1/2) and the second involves the large tumor suppressors LATS1 and LATS2 (LATS1/2), the adaptor Salvador homolog 1 (SAV1), and the MOB kinase activator (MOB1A-B). The transcriptional module associates the transcriptional co-activator with the PDZ-binding motif (TAZ) and the Yes-associated protein (YAP). Activation of the Hippo kinase cascade results in the phosphorylation of TAZ and YAP and facilitates their cytoplasmic retention and proteasomal degradation. However, when the pathway is inactive or its negative regulation on TAZ/YAP is inhibited, TAZ and YAP translocate to the nucleus to transcribe several target genes involved in cell growth, survival, migration, and self-renewal [68,85,86].

The identification of Hippo signaling as having biomarker potential could permit the development of strategies for signaling inhibition.

### 4.5. The JAK/STAT Pathway

Janus kinase signal transducers and activators of the transcription (JAK-STAT) signaling pathway are stimulated by cytokines. Compared with other signaling methods, the JAK-STAT pathway is relatively simple. Several studies have demonstrated that STAT signaling exerts an important influence on bCSC biology. Identification of the pathways that are altered in these cells is crucial for BC therapy. Accordingly, a study showed that STAT 3 activation is critical for the survival and proliferation of bCSCs. This investigation emphasized the importance of the development of therapeutic strategies focused on bCSC elimination [87].

In addition, it has been shown that the IL-6/JAK2/STAT3 pathway is significantly and preferentially activated in bCSCs with a basal-like phenotype in comparison with luminal-like tumor cell types. The inhibition of genes necessary for the growth and cellular proliferation of basal-like bCSC CD44^+^/CD24^−^ promotes a decrease in STAT activation. In addition, the inhibition of JAK2 diminishes the tumor number and growth. This type of study supports the importance of identifying the targets for the future development of more specific and effective BC therapies [88]. In contrast, it has been demonstrated that erythropoietin promotes tumorigenesis by activating JAK/STAT signaling in bCSCs, promoting their self-renewal and growth. Interestingly, hypoxia induces erythropoietin production in BC cell lines, but not in normal human mammary epithelial cells. Likewise, the use of pharmacologic JAK2 inhibition in addition to chemotherapy was shown to produce tumor growth inhibition in vivo [89]. Interestingly, it was reported that high ADAR1 expression was associated with immune responses. JAK/STAT signaling was found to be differentially regulated in basal-like tumors with high ADAR1 expression [90,91]. Adenosine deaminases acting on RNA (ADARs) catalyze the addition of adenosine-to-inosine (A-to-I) in double stranded RNA, a common post-transcriptional modification in mammals. Inosine is interpreted as guanosine during base-pairing, which can lead to codon changes, with the consequence of altered protein function, affecting the targeting and maturation of microRNAs. Thus, it has been shown in several studies that the transcriptomic as well as proteomic diversity introduced by A-to-I editing is exploited by tumor cells to promote cancer progression; for example, more editing events regulated by ADAR1 have been associated with cancer development, primarily due to the more abundant expression of ADAR1 in BC. The knockdown of ADAR1 attenuates proliferation and tumorigenesis, leading to robust translational repression [92,93,94]. Recently, ADAR1 and lncRNA interplay has been shown to play a role in cancer, suggesting that ADAR1 could alter the expression levels of lncRNAs, and these changes could be related to BC biology. Basically, the study showed that LINC00944 is responsive to ADAR1 up- and downregulation in breast cancer cells; LINC00944 expression has a strong relationship with immune signaling; and LINC00944 expression is correlated with the age at diagnosis, tumor size, and estrogen and progesterone receptor expression [95]. Additionally, ADAR enzymes modify a large proportion of cellular RNAs, contributing to transcriptome diversity and cancer evolution. It has been demonstrated that increased editing at the 3’UTR region occurs in breast cancer cells compared to in immortalized non-malignant cell lines, suggesting that there is a significant association between mRNA editing in genes related to cancer-relevant pathways and that ADAR1 plays a significant role in BC [96].

The main constituents are tyrosine kinase-related receptor, tyrosine kinase JAK, and the transcription factor STAT. The JAK protein family consists of four members (JAK1-3 and Tyk2) with seven JAK homology (JH) domains. The JH1 domain is the kinase domain, the JH2 domain is the “pseudo” kinase domain, and JH6 and JH7 are the receptor-binding domains. As its name infers, STAT (signal transducer and activator of transcription; STAT1, STAT2, STAT3, STAT4, STAT5a, STAT5b, and STAT6) plays a crucial role in signal transduction and transcriptional triggering. The structure of STAT is divided into an N-terminal conserved sequence, a DNA-binding region, an SH3 domain, an SH2 domain, and a C-terminal transcriptional activation region. When a signal is received, JAK activation occurs, catalyzing tyrosine phosphorylation of the receptor. The phosphorylated tyrosine on the receptor molecule can bind with the SH2 site of STAT. STAT binds to the receptor, producing tyrosine phosphorylation, which forms a dimer that promotes its arrival at the nucleus, affecting gene expression [63].

### 4.6. NF-κB Signaling Pathway

It is known that bCSCs overexpress components of the nuclear factor (NF)-κB signaling pathway, exhibiting high NIK expression and high NF-κB activity levels. Remarkably, NIK activity induces an increment in the bCSC CD44^+^/CD24^−^ population and enhances BC cell tumorigenicity. Interestingly, NIK inhibition reduces the expression levels of the *ALDH1A1*, *Nanog*, *Sox2*, and *Oct4* genes. This information supports the idea that NIK is an essential contributor to the phenotypic maintenance of bCSCs [97]. Moreover, through the NF-κB signaling pathway, the chemokine SDF-1 (stromal cell-derived factor 1) triggers the EMT in MCF-7 cells, which is conducive to obtaining the bCSC phenotype, contributing to cell migration and metastasis. These results suggest that SDF-1 is an attractive candidate for BC therapy [98].

NF-κB is a transcription factor that is rapidly inducible and encompasses five main proteins (p65, RelB, c-Rel, NF-κB1, and NF-κB2). The activity of this signaling pathway is regulated by two major pathways (canonical NF-κB signaling and non-canonical NF-κB signaling). In the canonical pathway, NF-κB activation occurs through the binding of ligands (e.g., bacterial cell components, IL-1β, TNF-α, or lipopolysaccharides) to their respective receptors, such as Toll-like receptors (TLRs), the TNF receptor (TNFR), the IL-1 receptor (IL-1R), and antigen receptors. Stimulation of these receptors leads to the phosphorylation and activation of IκB kinase (IKK) proteins, subsequently initiating the phosphorylation of IκB proteins. In the non-canonical pathway of NF-κB, receptors from different classes are involved, such as CD40, the receptor activator for NF-κB, B cell activation factor, TNFR2 and Fn14, and the lymphotoxin β-receptor. This pathway leads to the activation of NF-κB by inducing the kinase (NIK), which then phosphorylates and predominantly activates IKK1. The activity of the latter enzyme induces the phosphorylation of p100 to generate p52 [63].

### 4.7. PI3K/AKT/mTOR Signaling Pathway

Cancer cell survival is a fundamental mechanism for the proliferation and persistence of bCSCs. It is controlled by the canonical PI3K/AKT/mTOR signaling pathway, which is frequently activated in cancer. Phosphatidylinositol-3-kinase (PI3K) is an intracellular kinase. It is composed of the p85 regulatory subunit and p110 catalytic subunit, which have serine/threonine (Ser/Thr) kinase and phosphatidylinositol kinase activities. AKT, or protein kinase B (PKB), is a serine/threonine kinase that is expressed as three isoforms (AKT1, AKT2, and AKT3). AKT proteins are directly activated in response to PI3K, as well as being crucial effectors of PI3K. The mammalian target of rapamycin (mTOR) complex is a key gene target downstream of AKT. mTOR phosphorylates AKT, which leads to complete AKT activation [63]. The signaling pathways mediated by mTOR and PI3K are fundamental in the proliferation and cell survival of bCSCs. This supports their importance in bCSC maintenance and their role as attractive targets for the treatment of BC [87]. Furthermore, TGF-β is involved in the induction of CSCs in the bone microenvironment through the MAPK/ERK and AKT signaling pathways [99].

### 4.8. The TGF-β Pathway

It is known that TGF-β stimulates the malignant potential of mammary tumors in mice [100]. Recently, the participation of TGF-β in the generation of CSCs in the bone-microenvironment was evaluated. The results indicated that TGF-β signaling is involved in tumor growth and the tumor cell proliferation and that the bone microenvironment and TGF-β signaling are key factors that favor the development of a CSC niche, which can be responsible for chemoresistance [99]. Moreover, TGF-β signaling is activated in bCSCs, and its inhibition induces morphological changes that are characterized by transformation to an epithelial phenotype, which suggests that this mesenchymal appearance promoted by TGF-β signaling can be relevant to disease progression [101]. Within its pleiotropic functions, non-canonical TGF-β signaling regulates the expression of ILEI, which is an oncogene that is essential for the EMT in BC. Consequently, TGF-β activation or silencing of hnRNP E1 induces an EMT that enhances cell invasion and migration. hnRNP E1 knockdown significantly changes normal mammary epithelial cells into mesenchymal bCSCs in vitro and in vivo. Thus, the inhibition of ILEI protein levels results in diminished tumor growth. This represents a novel target for the development of BC therapies [102]. Another molecule that is up-regulated by TGF-β signaling is PMEPA1, which is a transmembrane protein induced by androgens. Depletion of this protein reduces the bCSC population. Contrarily, when it is overexpressed, bCSC production increases. Thereby, TGF-β contributes to bCSC maintenance [103]. For example, miR-181 is up-regulated at the post-transcriptional level by TGF-β, though miR-181 is capable of controlling the ataxia-telangiectasia mutated gene, which also is down-regulated by TGF-β. Both mechanisms of regulation induce an increase in mammospheres formation in bCSCs, and the knockdown of ataxia telangiectasia mutation enhances in vivo tumorigenesis, suggesting that the TGF-β pathway regulates bCSC development by interfering with this tumor suppressor [104].

TGF-β superfamily ligands bind to a type-II receptor that recruits a type-I receptor for phosphorylation. TGF-β superfamily ligands include bone morphogenetic proteins (BMP), growth and differentiation factors (GDFs), anti-Mullerian hormone (AMH), Nodal, and TGF-β. These ligands are divided into two groups: TGF-β/activin and BMP/GDF. Meanwhile, Smad proteins are divided into three subfamilies: receptor-activated or pathway-restricted Smad (R-Smads), Co-Smad, and inhibitory Smad (I-Smads). In summary, the type-I receptor phosphorylates receptor Smads (R-Smads), which bind to the common pathway Smad (co-Smad). The R-Smad/co-Smad complex acts as a transcription factor in the nucleus to regulate the expression of target genes [38,63].

In summary, the analysis of these signaling pathways has helped us to understand several of the mechanisms involved in the development and regulation of bCSCs. In fact, several studies have demonstrated the relevance of these signaling pathways in the biology of bCSCs, showing that they represent attractive targets for the design of drugs to eliminate BC.

## 5. The Prominent Role of miRNAs in Breast Cancer Stem Cells

miRNAs are small, non-coding, single-stranded RNA molecules with a length of ~19–25 nucleotides that are evolutionarily conserved and operate as post-transcriptional regulators of gene expression [105,106]. Their function is based on binding to complementary target sequences in messenger RNA (mRNA) and interfering with the translational machinery, thereby preventing or altering the production of the protein product (repressing translation). The binding of miRNA to its target, mRNA, also triggers the recruitment and association of mRNA decay factors, leading to mRNA destabilization and degradation [106]. These molecules were discovered in 1993, and data demonstrate that Lin-4 regulates the transcription of Lin-14, impacting significantly on the development of *Caenorhabditis elegans* [107]. Since its discovery, from recognition of its existence and importance in several mechanisms, hundreds of investigations have been carried out to identify the role of miRNA in various pathologies, including BC. In recent years, the study of miRNAs has increased substantially, due to their ubiquitous involvement in several cellular pathways. In BC research, they have been shown to act as modulators of gene expression, displaying both tumor-promoting and tumor-suppressive functions. Accordingly, their association with several carcinogenic processes that contribute to poor survival, high susceptibility, and progression has been found. Indeed, several miRNAs have been classified as biomarkers of recognition, progression, and prognosis of disease [105]. miRNAs regulate numerous mechanisms, functions, and features of bCSCs through regulation of the expression of promoter (oncogenes) or tumor suppressor genes. The central impact is related to the self-renewal, differentiation, invasion, metastasis, and therapy resistance of bCSCs. In this sense, currently, focused and directed therapeutic approaches based on miRNAs are being developed against one or several proteins involved in these signaling pathways (Figure 1).

In the next section, we will describe the involvement of miRNAs in the regulation of several signaling pathways of bCSCs (Table 2, Figure 1). The understanding of cellular regulation through miRNAs has therapeutic potential that could be exploited to develop better strategies for BC.

The current evidences indicate that reduced expression of the Let-7 family is associated with poor outcomes in patients with BC. Thus, it is considered to be a fundamental tumor suppressor that targets several oncogenic mRNAs [113,131,132]. Wnt signaling in a β-catenin dependent manner regulates the expression of Let-7 by suppressing its mature forms through Lin28 up-regulation and being the main negative regulator of Let-7 biogenesis. Let-7 participates in the expansion control and self-renewal of bCSCs [73]. Additionally, high expression of Let-7 b-c is correlated with better prognosis in patients with ERα^+^ BC. Data suggest that Let-7c inhibits estrogen-induced self-renewal of ERα^+^ bCSCs and decreases ERα expression, inhibiting estrogen activation through Wnt signaling [112]. On the other hand, miR-600 possesses tumor suppressor capabilities. The overexpression of this factor reduces bCSC self-renewal through the inhibition of Wnt by suppressing the expression of stearoyl desaturase 1 (SCD1) and diminishing in vivo tumorigenicity. Its silencing promotes bCSC expansion, and Wnt signaling activation promotes self-renewal [115]. MiR-142 is another oncomiR with an important contribution to malignant bCSC behaviors. Its main target is the tumor suppressor gene APC (Adenomatous polyposis coli) that is present in both normal and malignant mammary cells. MiR-142 is able to down-regulate APC by suppressing the translation of APC mRNA. This suppression promotes activation of the canonical Wnt/β-catenin signaling pathway, although aberrant proliferation capability and cellular apoptosis in bCSCs are regulated through miR-142 [116]. In contrast, MiR-146a is another miRNA that is implicated as having a tumor-suppressive role as it promotes the asymmetric division of bCSCs. Mechanistically, miR-146 degrades Lin28, a blocker of maturation of the Let-7 family; Let-7c was previously revealed to have suppressive functions related to SC expansion. Additionally, Let-7 controls Wnt signaling pathway activity and could be strengthened due to the miR-146 inhibition of H19, forming a positive feedback regulation loop with Let-7. This miR-146a/Lin28/Wnt axis has been implicated in the prevention of symmetric cell division and the prohibition of bCSC expansion [114]. However, the overexpression of miR-208a promotes the high proportion and self-renewal of ALDH1^+^ bCSC cells. Likewise, it enhances the upregulation of Lin28 through the stimulation of both Sox2 and β-catenin, producing the inhibition of Let-7a [113].

MiR-204 is another miRNA that inhibits the self-renewal of bCSCs through the suppression of Sam68. However, studies have established that miR-204 is commonly down-regulated in BC [117]. Sam68 is a Src-associated protein that is involved in intracellular signal transduction, proliferation, and apoptosis in mitosis. Its up-regulation is correlated with self-renewal, and its forced downregulation inhibits the proliferation and tumorigenicity of breast cancer cells [133]. Evidence suggests that Sam68 promotes self-renewal potential, and thus it could be a novel therapeutic target for BC [117]. It has been shown that NIMA-related kinase 2 (NEK2) is a target of miR-128-3p, and it has been found to be up-regulated in BC. Although miR-128-3p is commonly downregulated in breast cancer cells, its overexpression contributes to the inhibition of cell proliferation, migration, invasion, and self-renewal of bCSCs in vitro as well as tumorigenicity in vivo. These effects are triggered because miR-128-3p promotes the inhibition of the Wnt pathway through the downregulation of NEK2. Both NEK2 and miR-128-3p contribute to BC development, making them attractive targets for treatment [134].

In addition, miR-1 expression has been inversely associated with BC development. miR-1 over-expression reduces the number of SKBR3/CSCs. However, its inhibition augments the number of MCF-7/CSCs. An in vitro analysis showed that miR-1 inhibits the proliferation, stemness, and migration promoted by CSCs. Augmentation of miR-1 expression inhibits the growth of MCF-7/CSCs, while miR-1 inhibition promotes the growth of MCF-7/CSCs in vivo. Moreover, miR-1 promotes their modulation capacity through binding to Frizzled 7 and TNKS2 and inhibiting Wnt/β-catenin signaling [118]. Similarly, FSTL1 increases oncogenesis in BC by enhancing stemness and chemoresistance via Wnt/β-catenin signaling through integrin β3. MiR-137 reduces the FSTL1 mRNA and protein levels, regulating FSTL1 activity. These findings indicate that the miR-137/FSTL1/integrin β3/Wnt/β-catenin signaling axis regulates stemness and chemoresistance [119].

miR-31 expression is directly activated by the NF-κB pathway in mammary tissue. In breast tumors, miR-31 was also found to be markedly up-regulated; however, its downregulation resulted in a reduction of the bCSC subpopulation and decreased tumor initiation and metastasis abilities. Data also show that miR-31 mainly up-regulates Wnt/β-catenin activity by targeting Xin1, Gsk3β, and Dkk1 and suppresses TGF-β signaling through the regulation of targets such as Smad3 and Smad4 in the mammary epithelium [120].

miRNAs play pivotal roles in the regulation of signaling pathways that modulate the stemness of bCSCs. The Notch4 signaling pathway was found to be negatively regulated by the action of miR-34c in bCSCs, inducing the suppression of self-renewal ability, the EMT program, and cell migration and proliferation in bCSCs [108]. Overexpression of miR-34a was shown to suppress BC stemness in vitro and increase chemosensitivity by directly targeting Notch1. In contrast, a reduced expression of miR-34a plays an essential role in the initiation, progression, and metastasis in BC [109,110]. Studies supports the role of Notch signaling in chemoresistance, as it has been demonstrated that Jagge2 is highly expressed in triple-negative breast cancer cells that are resistant to paclitaxel. Likewise, their expression is associated with bCSC features. This was corroborated by the performance of Jagged2 knockdown which inhibited bCSC properties and chemoresistance. Interestingly, miR-200 knockdown triggered the ability to revert this mechanism [111].

The miR-221/222 cluster functions as an oncogene as it enhances the growth, migration, and invasion of breast cancer cells. Investigations have shown that the tumor suppressor phosphatase and tensin homolog (PTEN) is a target of miR-221. Its mechanism of action through exerting the downregulation of PTEN occurs through an increase in AKT phosphorylation. Thus, cell migration, proliferation, invasion, and self-renewal are the principal hallmarks of MCF-7 cell modulation via targeting the PTEN/AKT pathway [125]. Notably, miR-221/222 is overexpressed in the aggressive bCSC subpopulation from MDA-MB-231 cells, and the mammosphere formation capacity increases after the ectopic expression of miR221/222 or PTEN knockdown. Further, the participation of the AKT/NF-κB/COX-2 pathway in the dissemination of bCSCs was corroborated in xenografted tumors, where increases in AKT phosphorylation, NF-κB p65, and phosphorylated p65, and cyclooxygenase-2 were found [124].

MiR-21 plays an important role in the migration and invasion of BC. It has been proposed that miR-21 is responsible for acquisition of the EMT phenotype, allowing migration and invasion in bCSC MCF-7 [135]. Using an inhibitory approach, the antagomir of miR-21 induces the reversion of EMT and the bCSC phenotype in MDA-MB-231 cells, promoting the inactivation of AKT/ERK1/2 and PTEN up-regulation. Moreover, inhibition of the ERK1/2 and PI3K-AKT pathways suppresses the EMT and the CSC phenotype, indicating the participation of these pathways in the ability of miR-21 to regulate these hallmarks in bCSCs [126]. The metastamiR miR-10b promotes the metastasis and migration of bCSCs; thus, their overexpression in MCF-7 cells induces high self-renewal and the expression of stemness and EMT markers. Similarly, PTEN is a potential target of miR-10b. miR depletion induces the up-regulation of PTEN and promotes a decrement in AKT activity. These data indicate that miR-10b plays an essential role in regulating the self-renewal and migration of the bCSC phenotype by modulating the PTEN/PI3K/AKT pathway [127].

Another molecular target of miRNAs is the programmed cell death ligand 1 (PD-L1), an immune checkpoint molecule. PD-L1 expression is associated with EMT and CSC markers; in addition, overexpression of PD-L1 in BC may promote the maintenance of stemness in bCSCs [136]. Moreover, the overexpression of PD-L1 in tissues and cells promotes chemoresistance and enhances the stemness of bCSCs through PI3K/AKT and ERK1/2 signaling activation. The direct action of miR-873 via targeting and inactivating downstream PI3K/AKT and ERK1/2 signaling attenuates the stemness and chemoresistance abilities of bCSCs in vivo and in vitro [128].

Hippo signaling is associated with BC stemness and maintenance and is modulated by miRNAs. Thus, a member of the miR-302/372/373/520 family, miR-520, has been reported as acting as an oncogene in BC. MiR-520b is up-regulated in BC tissue and bCSCs, indicating that its expression is associated with poor prognosis in patients. LATS2, a key constituent of Hippo/Yap signaling, is predicted to be a potential target of miR-520b. Additionally, the expression of LATS2 is significantly downregulated in BC tissue. The mRNA levels and phosphorylation of YAP are also significantly decreased in BC. This strongly suggests that miR-520b promote the maintenance of bCSC stemness through the LATS2 target through Hippo/YAP signaling activation [121].

Depending on the conditions, interestingly, miR-93 also controls the proliferation and differentiation of normal MaSCs by acting as a tumor suppressor. Further, it was demonstrated that miR-93 modulates the fate of bCSCs by regulating their proliferation and differentiation capabilities. Their overexpression decreases bCSCs in vivo (mouse xenograft model) and in vitro (cell lines). Further, miR-93 induces the MET phenotype and CSC depletion, having a strong association with the downregulation of TGF-β, JAK1/STAT3, and AKT3 signaling [122].

miR-141 expression is regulated through progesterone signaling in BC cells. Moreover, miR-141 inhibition is involved in cell-dedifferentiation mediated by progesterone. This suppression leads to increased CD44^high^ and CK5^+^ populations, as well as PR expression. Stat5 is a predicted target of miR-141 and specifically regulates Stat5a in BC cells. Stat5 inhibition diminishes the populations of CD44^high^ and CK5^+^ cells stimulated by progesterone [123].

Certainly, TGF-β signaling plays a fundamental role in cancer development by modulating diverse biological events including cell proliferation, cell migration, and cell differentiation through regulation of the activation of several enzymes. Indeed, it exerts a bimodal function, whereby the TGF-β pathway acts as a tumor suppressor and as a tumor promoter depending on the stage of cancer progression [137]. In BC, it has been demonstrated that the TGF-β pathway is involved in both supporting the growth and survival of cells as well as participating in cell apoptosis induction [138,139]. In the murine mammary cell line NMuMG, the obstruction of autocrine TGF-β signaling reduces the capability of cells to grow anchorage-independently and to resist apoptosis induction. Interestingly, the abolition of TGF-β signaling promotes the attenuation of the EMT, mammosphere formation, and the expression of stem cell markers, supporting the idea that autocrine TGF-β signaling is involved in the maintenance and survival of murine bCSCs [140]. On the other hand, the exposition to TGF-β increases the bCSC population with mammosphere formation ability. This mechanism is modulated by miR-181, which is up-regulated by TGF-β. Indeed, miR-1 overexpression or the elimination of its target ataxia-telangiectasia mutation induces mammosphere formation in BC cells. These data suggest that a mechanism involving the TGF-β pathway regulates CSC properties by interfering with mutated ataxia-telangiectasia [104].

Currently, due to bCSC chemoresistance, the search for therapeutic alternatives is an area of great exploitation and interest. Investigation of caffeic acid, a hydroxycinnamic acid (3,4-dihydroxycinnamic acid, CaA), supports its potential as an anti-tumoral molecule with the ability to inhibit the proliferation, migration, and invasion processes of human cancer cells. CaA attenuates bCSC properties, such as decreased formation of mammospheres and the expression of stem cell markers or genes related-stemness, improving the expression of miR-148a but inhibiting the expression of Smad2. The mechanism of CaA, whereby it enhances miR-148a expression, is based on the inhibition of DNA methylation in its promoter. miR-148a mediates the inhibition of TGF-β/SMAD2 signaling due to *Smad2*, the direct target gene. Knockdown of miR-148a blocks CaA-induced decreased expression of Smad2. Overexpression of miR-148a by itself without CaA treatment decreases the Smad2 level. These results suggest that CaA inhibits Smad2 activation by miR-148a in bCSCs [129]. Another molecule with anti-cancer properties that has been evaluated is the phytochemical Glabridin (GLA), which exerts a similar mechanism to CaA. Similarly, GLA attenuates bCSC properties through the miR-148a/TGF-β/SMAD2 signal pathway in vitro and in vivo [130].

As previously described, miRNAs mechanistically in a bimodal way depending on the various molecular mechanisms involved, significantly contributing to the control or promotion of malignant bCSC behaviors. Better knowledge is needed, as the development of therapeutic strategies against miRNAs might represent a novel strategy for BC treatment.

## 6. The lncRNA: Key Molecules in the Breast Cancer Stem Cells

LncRNAs constitute another important class of non-coding RNAs with lengths exceeding 200 nucleotides; as a particular characteristic, they are not translated into proteins [141]. They regulate a diverse range of genes at the epigenetic, transcriptional, and post-transcriptional levels through interactions with miRNAs, mRNAs, proteins, and genomic DNA. Thus, lncRNAs are actively involved in chromatin rearrangement, histone modification, the regulation of transcription, the regulation of basal transcription machinery, post-transcriptional regulation splicing translation, and the regulation of gene expression. Their association with carcinogenesis is based on their aberrant expression in several cancer types, including BC. LncRNAs influence the modulation of stemness properties, as well as the growth and apoptosis of cancer cells. The main lncRNAs described as having important associations with BC are MALAT-1, HOTAIR, and H19, which regulate the fundamental signaling pathways involved in oncogenic and tumor suppression [141,142]. The study of the mechanisms that have been found to be deregulated has contributed to improved understanding of the potential of lncRNAs to act as biomarkers in diagnosis, prognosis, and targeting in BC. The direct contribution of lncRNAs is linked with their roles in tumorigenesis, apoptosis, metastasis, chemoresistance, radioresistance, and angiogenesis through various essential signaling pathways [141,143,144]. Fascinatingly, lncRNAs play transcendent roles in bCSC biology by acting as sponges of miRNAs. In this way, the lncRNAs influence the expression levels of the targets of miRNA. In this regard, several studies have reported the mechanistic process of regulation mediated by lncRNAs (Table 3).

To demonstrate the protagonist role of lncRNAs as regulatory molecules that participate in the acquisition of tumorigenic phenotypes through their association with quiescent SC populations, a study established the oncogenic association of an uncharacterized lncRNA named HAL with the tumorigenic phenotype of MCF-7-MCTS under hypoxic conditions. LncRNA-HAL is overexpressed in quiescent SC MCF-7-MCTS. Its knockdown is associated with cell proliferation, migration, and survival mechanisms; thus, HAL silencing increases cell proliferation and impairs the proportion and function of CSCs, resulting in decreased tumor implantation in vivo [145]. lncRNA-LINC01133 is another lncRNA that is regulated by the microenvironment and is mainly orchestrated by mesenchymal stem/stromal cells (MSCs). An investigation supported the idea that lncRNA-LINC01133 stimulates the bCSC phenotype in triple-negative BC and acts as a malignancy indicator. LncRNA-LINC01133 is strongly induced by MSCs in BC cells. The main contribution of LINC01133 is to induce the bCSC phenotype and activate its principal features. In addition, it has been found that is a direct mediator of the triggering of the miR-199a-FOXP2 axis by MSCs in BC cells and a critical regulator of the pluripotency, representing a biomarker and prognosticator of disease [146].

Recently, it has been demonstrated that lncRNA-Hh acts as a strategic protagonist by facilitating Twist-induced bCSC properties by regulating the Hedgehog signaling pathway. An enhancer of hedgehog signaling, GAS1, is the direct target of lncRNA-Hh. This induces the up-regulation of Sox2 and Oct4 expression, promoting the acquisition of stemness properties. This study proposed a new target in bCSCs [147]. Moreover, the lncRNA SOX21-AS1 is up-regulated in BC tissue and is associated with poor prognosis. Remarkably, SOX21-AS1 knockdown was shown to reduce proliferation, invasion, tumor growth, stem factor expression of the CD44^+^/CD24^−^ population in vitro and in vivo. This investigation suggests that SOX21-AS1 controls bCSC properties and carcinogenesis via targeting Sox2 [148]. Posteriorly, it was corroborated that the lncRNA SOX21-AS1 is highly expressed in bCSCs of MCF-7 and MDA-MB-231, promoting bCSC properties as well as the proliferation, migration, and invasion abilities of CSC-MCF-7 cells through inhibiting the Hippo signaling pathway [149].

On the other hand, lncRNA THOR plays oncogenic roles in several tumors, including BC. The investigation supports the overexpression of THOR in samples of triple-negative BC. Interestingly, the silencing of THOR induces reductions in mammosphere formation, stemness marker expression, and ALDH1 activity of bCSC. Mechanically, THOR directly binds to β-catenin mRNA, increasing its stability and expression and assisting bCSC stemness through activating β-catenin signaling [150]. LncCCAT1 (colon cancer-associated transcript-1) is significantly up-regulated in BC tissue and bCSCs, leading to poor patient outcomes. Mechanistically, it has been suggested that LncCCAT1 can interact with miR-204/211, miR-148a/152, and Annexin A2, promoting the up-regulation of T-cell factor 4 by competitive binding to miR-204/211 or the translocation of β-catenin to the nucleus through interaction with miR-148a/152 and Annexin A2, leading to Wnt pathway activation and stimulating the proliferation, stemness, migration, and invasion of bCSCs [151]. LncRNA LUCAT1 (lung cancer-associated transcript 1) regulates stemness features in BC through the Wnt/β-catenin pathway. It has been reported that LUCAT1 is expressed in BC tissue and highly expressed in bCSCs. Its expression is correlated with tumor size and lymph node metastasis. Likewise, its high expression is associated with poor prognosis and low survival due to the promotion of bCSC proliferation. In addition, LUCAT1 and the transcription factor TCF7L2 are targets of miR-5582-3p, LUCAT1 functions as a miR-5582-3p sponge, enhancing the Wnt/β-catenin signaling pathway [152]. In addition, in BC, lncRNA H19 is aberrantly up-regulated and functionally associated with numerous biological mechanisms, such as cell proliferation, invasion, and tumoral apoptosis [144]. In addition, lncRNA H19 is highly expressed in bCSCs and breast tumor samples. Ectopic overexpression of H19 enhances clonogenicity, migration, and mammosphere formation. Conversely, these features are repressed when H19 is silenced. The mechanism of H19 function involves competing with endogenous RNA to sponge miRNA Let-7, blocking the bioactivity of Let-7, an upstream repressor of H19 [153]. Furthermore, H19 has been related to a poor prognosis in BC patients and the promotion of cancer stemness. It was demonstrated that the inhibition of H19 reduces mammosphere formation in HCC1934 and iCSCL10A cell lines [154]. In addition, H19 is stimulated by the control of the symmetric division of bCSCs, increasing self-renewal and acting as a sponge that inhibits Let-7c availability, thereby impacting Wnt activation. This study suggests that there is a double-negative feedback loop between sponge H19 and targeted Let-7c through estrogen activated Wnt signaling, which is fundamental for bCSC division [155].

Homeobox transcript antisense RNA (HOTAIR) is related to the development and metastasis of BC [144]. HOTAIR is highly up-regulated in bCSC MCF7 and MB-231 and induces modulation of the proliferation, migration, and self-renewal abilities of bCSCs. Furthermore, an analysis indicated that HOTAIR regulates the transcription of miR-34a, which leads to Sox2 up-regulation [156]. HOTAIR is able to down-regulate miR-7 through modulation of the expression of HoxD10 in the bCSCs of MCF-7 and MDA-MB-231 cells. In turn, miR-7 acts as a tumor suppressor, inhibiting invasion and metastasis in NOD/SCID mice and diminishing the bCSC population. Interestingly, also it is involved in the reversion of the EMT phenotype in MDA-MB-231 cells by targeting the oncogene SETDB1. Stat3 suppression is the result of the repression of SETDB1 by miR-7, which impacts the downregulation of c-Myc and twist [157].

Nuclear lncRNA metastasis-associated lung adenocarcinoma transcript 1 (MALAT1) is the most conserved and highly abundant lncRNA in normal tissues, suggesting that it has vital biological implications [144]. However, also it has been associated with malignant comportment, which performs a critical role in stemness maintenance in several CSCs. MALAT1 is overexpressed in bCSC MCF7, and its knockdown considerably decreases the proportion of bCSC MCF7 and mammosphere formation in vitro. MALAT-1 affects bCSCs through the regulation of Sox-2 and may serve as a novel biomarker for predicting malignancy and as a potential therapeutic target for breast tumor metastasis [158]. In fact, lncRNA FEZF1-AS1 is also up-regulated in BC and is associated with poor prognosis in patients. LncRNA FEZF1-AS knockdown reduces the ability to form mammospheres, the expression of stem cell markers, and the rate of CD44^+^/CD24^−^ production. In addition, growth, proliferation, migration, and invasion are significantly inhibited. An analysis demonstrated that FEZF1-AS1 modulates bCSC and Nanog expression through sponging miR-30a [160]. Further, an exploration of its expression showed that lncRNA LINC00511 is highly expressed in BC, which is correlated with the poor prognosis of patients. Mechanistically, LINC00511 functions as a miR-185-3p sponge and targets E2F1, promoting the transcriptional regulation of Nanog. Accordingly, the LINC00511/miR-185-3p/E2F1/Nanog alliance facilitates BC stemness and the tumorigenesis of bCSCs [161]. Recently, LncRNA FGF13-AS1 was identified as one of the downregulated lncRNAs in BC. A functional analysis showed that FGF13-AS1 functions as a tumor suppressor to inhibit BC cell proliferation, migration, and invasion by impairing glycolysis and stemness properties. Through a reduction of the half-life of c-Myc mRNA and inhibition of the association of IGF2BPs and Myc mRNA, Myc inhibits FGF13-AS1. The FGF13-AS1/IGF2BPs/Myc feedback loop represents a promising novel therapeutic target for BC patients [159]. Finally, LINC00617 is up-regulated in BC samples and functions as an important regulator of EMT, promoting the progression and metastasis via the activation of Sox2 [162].

Further studies on lncRNAs are fundamental to increase the understanding of the complex networks involved in bCSC biology. These investigations will provide information about the mechanisms of control of these molecules in bCSCs (Figure 2), allowing us to comprehend the mechanisms by which bCSCs induce self-renewal, propagation, and chemoresistance. In this regard, lncRNAs are excellent therapeutic targets, because they are stable and easily detectable in breast tissues.

## 7. Strategies of Elimination or Resistance in bCSC: microRNA and lncRNAs as Protagonist

As it has been described, bCSC play roles in metastasis, resistance, and recurrence affecting anticancer therapy efficacy. Targeting of bCSC using microRNAs, lncRNAs or associated pathways become a new approach for the potential treatment of BC. Currently, the use of natural compounds is a novel approach for cancer therapy.

Tangeretin inhibited cell proliferation, tumor growth, and modestly induced apoptosis in CSC. Furthermore, the tangeretin inhibits the Stat3 signaling pathway and induces CSC death, indicating that tangeretin may be a potential natural compound that targets BC cells and CSC [163]. It is well known that Notch signaling mediates BC chemoresistance. In this concern, the high expression of Notch1 stimulates the reduction of the miR34 tumor-suppressor in MCF7/ADR cells. Ectopic miR34a expression reduced bCSC stemness properties, and enhance sensitivity to doxorubicin treatment by directly targeting Notch1. This approach represents a novel therapeutic strategy for chemoresistant BC [110].

Moreover, CaA inhibits the bCSC properties via miR-148a mediated inhibition of TGF-β-SMAD2 signaling pathway both in vitro and in vivo. CaA enhanced the expression of miR-148a by inducing DNA methylation [129].

Glabridin exhibited effective antitumor properties in various human cancer cells. Glabridin attenuated the bCSC properties through miR 148a/TGF-β/SMAD2 signal pathway in vitro and in vivo [130]. The microRNA Let-7a was previously identified to target stem-like cells in cancer via inhibition of ERα, the suppressive effects exerted by Let-7 on stem-like cells involved Let-7c/ER/Wnt signaling, and the functions of Let-7c exerted with tamoxifen were dependent on ER. Taken together, the findings identified a biochemical and functional link between Let-7 and endocrine therapy in bCSC [164]. Other study, used let-7a as therapeutic agents for the regulation of bCSC as well as BC when they are formulated in Herceptin-conjugated cationic immuno-liposome with hyaluronic acid and protamine, this efficient liposomal delivery system for the combination of miRNA and siRNA to target the bCSC can be used as an efficacious therapeutic modality for breast cancer [165]. Furthermore, 6-methoxymellein inhibits the proliferation and migration of BC cells, reduces mammosphere growth, and decease CD44^+^/CD24^−^ population as well as, diminish the expression of c-Myc, Sox-2 and Oct4. 6-Methoxymellein reduces the nuclear localization of NF-κB subunit p65/p50 and secretion of IL-6 and IL-8 [166]. In addition, TV-circRGPD6 suppresses bCSC metastasis via the miR-26b/YAF2 axis. TV-circRGPD6 nanoparticle that selectively expresses circRGPD6 in metastatic bCSC to eradicate breast cancer metastasis, therefore providing a novel avenue to treat BC [167]. Plumbagin exerted anti-cancer activity and eliminated stem-like properties by attenuating Wnt/β-catenin signaling. Suggesting that Plumbagin could be a promising agent to treat endocrine resistant breast cancer [168].

Jagged2 maintains bCSC properties and paclitaxel resistance via regulating miR-200 [111]. Linc-ROR also is an important marker for multidrug resistance in BC. Linc-ROR overexpression decreased sensibility of 5-FU and paclitaxel with diminish of E-cadherin expression, increase of Vimentin, N-cadherin expression [169]. Contrarily, LINC00968 overexpression contributes to reduced drug resistance in BC cells by inhibiting the activation of the Wnt2/β-catenin signaling pathway through silencing Wnt2 [170].

## 8. Clinical Trials

The National Library of Medicine contains a report on the clinical trial NCT01231386, which supports the fundamental role of miRNAs, implicating their dysregulation in BC tumorigenicity as a way of identifying miRNA markers of prognosis and their use as indicators and potential targets for personalized therapies. In this proposal, specimens from patients treated in a clinical BC program who are on already existing protocols (IRB 05,091 and 05015) will be characterized by Dr. Rossi and collaborators, and the information gained will be applied to develop specific therapies. Further, another clinical trial identified as NCT03779022 showed that serum microRNA expression could be used as an early marker for determining the BC risk. This validates the idea that aberrant miRNA expression in human BC is correlated with tumor development, progression, and drug resistance.

## 9. Closing Remarks

Several studies support the idea that bCSCs are responsible for the propagation, chemoresistance, and recurrence of BC, thus confirming a prominent role in cancer pathogenesis and providing an attractive area of research. The development of personalized medicine targeting effectively the bCSCs may in a future, improve the outcomes of BC patients.

It has been demonstrated that miRNAs and lncRNAs influence processes involved in cancer development, such as tumorigenesis, apoptosis, metastasis, chemoresistance, radioresistance, and angiogenesis, and are critical regulatory elements in the epigenetic mechanisms bCSC biology and plasticity regulation. It has been suggested that they are biomarkers that could be used in cancer diagnosis and prognosis markers in patients with BC. Clinical trials support the idea that miRNA expression can act as a biomarker for early detection and therapeutic response monitoring in BC patients. LncRNAs and miRNAs represent attractive molecular targets and molecular tools that might be used for bCSC directed therapy. Further studies are necessary to increase the understanding of the complex networks involved in controlling SC modulated by miRNAs and lncRNAs. These investigations will allow the discovery of molecular mechanisms corrupted by differentiated cancer cells and bCSCs that promote BC development. Although much remains to be understood, future studies on the mechanisms, expression, and targets that miRNAs and lncRNAs modulate will provide us with information on the mechanisms that these molecules control in bCSCs. The development of adjuvant therapies using standard treatments, lncRNA/miRNAs, and natural compound could be a promising strategy to treat BC.

## Figures and Tables

**Figure 1 biomolecules-11-00380-f001:**
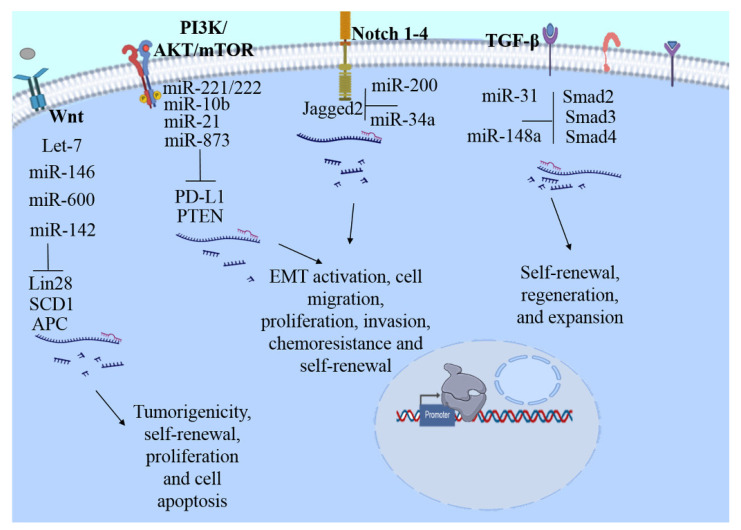
Main signaling pathways modulated by miRNAs in bCSC. Stem cell maintenance is complex and highly regulated processes, several pathways are essential for the maintenance of bCSC, including NF-κB, PI3K/AKT/mTOR, Notch, Hedgehog, Wnt, Hippo, JAK/STAT, TGF-β, and signaling estrogen receptor-mediated. Deregulation of these pathways can be one of the main reasons for the appearance or exacerbation of the main hallmarks such as maintenance, self-renewal, tumor resistance, relapse, recurrence, aggressiveness, and metastasis of bCSC, among others. Some targets modulated by miRNAs: Wnt, stearoyl desaturase 1 (SCD1), Adenomatous polyposis coli (APC), Lin28; Notch, Jagged2; PI3K//AKT/mTOR, phosphatase and tensin homolog (PTEN) programmed cell death ligand 1 (PD-L1), and TGF-β, SMAD 2-4.

**Figure 2 biomolecules-11-00380-f002:**
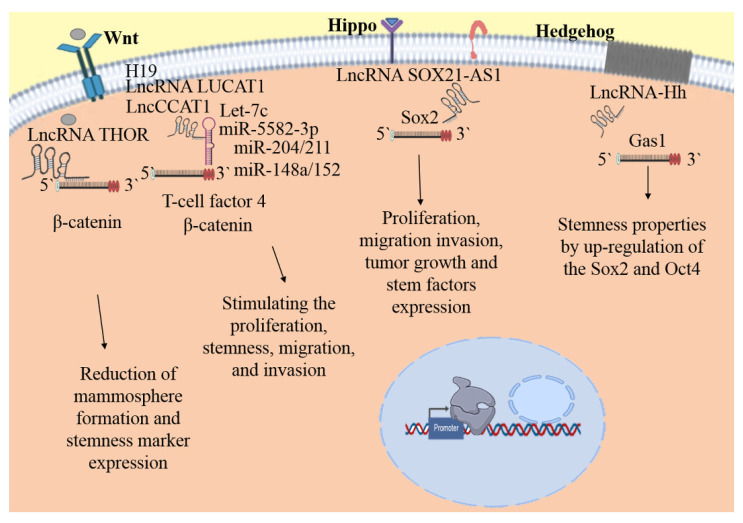
Signaling pathways modulate by lncRNAs in bCSC. The lncRNAs playing transcendent roles in the bCSC biology acting as sponges of miRNAs, the lncRNAs influence the expression levels of the targets of miRNA, or by direct bind to mRNA, impacting in the proliferation, maintenance, self-renewal, migration, and invasion of bCSC. Until now, the information about the lncRNA and the signaling pathways is scarce. Wnt, Hippo, and Hh are the main explored pathways related with lncRNAs regulation in bCSC. Some targets modulated by miRNAs: Wnt, β-catenin, T-cell factor 4; Hippo, Sox2 and Hh, Gas1.

**Table 1 biomolecules-11-00380-t001:** Stem cells markers used for the bCSC designation and isolation.

Marker	Protein	Reference
CD44	Cell adhesion molecule	[46,47]
CD24	Cell adhesion molecule	[45,46,47]
ALDH1	Aldehyde dehydrogenase 1	[46,47,51]
CD133	Prominin 1	[52,53,54]
CD49f	α6-integrin	[55,56]
CD90	GPI-anchored glycoprotein	[57,58]

**Table 2 biomolecules-11-00380-t002:** MiRNAs implicate in the bCSC biology.

miRNA	Target	Hallmarks	Reference
miR-34c	Not apply	Self-renewal, EMT	[108]
miR-34a	Not apply	Stemness and chemoresistance	[109,110]
miR-200	Not apply	Stemness and Chemoresistance	[111]
Let-7c	Not apply	Self-renewal and tumorigenicity	[112]
miR-208a-Let-7	Lin28, Sox2 and β-catenin	Self-renewal	[113]
Let-7	Lin28	Expansion and self-renewal	[73]
miR-146a-Let-7c	Lin28	Asymmetric division and Expansion	[114]
miR-600	SCD1	Self-renewal and Expansion	[115]
miR-142	APC	Proliferation and apoptosis	[116]
miR-204	Sam68	Self-renewal	[117]
miR-1	Frizzled 7 and TNKS2	Proliferation and migration	[118]
miR-137	FSTL1	Chemoresistance, proliferation, and stemness	[119]
miR-31	Xin1, Gsk3β, Dkk1, Smad3 and Smad4	Self-renewal, maintenance and metastasis	[120]
miR-520b	LATS2	Stemness maintenance	[121]
miR-93	Not apply	Proliferation and differentiation	[122]
miR-141	STAT5A	Proliferation and expansion	[123]
miR-221/222	PTEN	Self-renewal Stemness properties and tumor growth	[124,125]
miR-21	PTEN	EMT and Stemness properties	[126]
miR-10b	PTEN	EMT, self-renewal metastasis and migration	[127]
miR-873	PD-L1	Stemness and chemoresistance	[128]
miR-181	ATM	Mammospheres Formation	[104]
miR-148a	SMAD2	Proliferation, migration, invasion processes and mammospheres Formation	[129,130]

**Table 3 biomolecules-11-00380-t003:** Role of lncRNAs as gene regulators in bCSC.

LncRNA	miRNA	Target (mRNA)	Hallmarks	Reference
HAL	Not apply	Not apply	Stemness, proliferation, migration, and cell survival	[145]
LINC01133	miR-199a	KLF4 and FOXP2	Stemness and growth	[146]
HH	Not apply	GAS1	Stemness maintenance, mammospheres formation, EMT and self-renewal	[147]
SOX21-AS1	miR-429	SOX2	Proliferation, invasion, tumor growth, stem factors expression	[148]
SOX21-AS1	Not apply	Not apply	Stemness, proliferation, migration and invasion abilities	[149]
THOR	Not apply	β-Catenin	Mammospheres formation and stemness marker expression	[150]
CCAT1	miR-204/211, 148a/152	TCF4	Proliferation, stemness, migration, and invasion	[151]
LUCAT1	miR-5582-3p	TCF7L2	Stemness features, proliferation and tumor growth	[152]
H19	Let-7	Lin28	Stemness maintenance, clonogenicity, migration, mammospheres formation, and tumor growth	[153]
H19	miR-103, 107, let-7, and 29b-1,	Not apply	Stemness, mammospheres formation and tumor size	[154]
H19	Let-7c	Not apply	Symmetric division and self-renewal	[155]
HOTAIR	miR-34a	Sox2	Proliferation, invasion, Self-Renewal, Tumor Formation and Migration	[156]
HOTAIR	miR-7	SETDB1	Invasion, metastasis and EMT	[157]
MALAT-1	Not apply	Sox2	Stemness maintenance, mammospheres formation, proliferation, migration, and invasion	[158]
FGF13-AS1	Not apply	c-Myc and IGF2BP	proliferation, migration, invasion, glycolysis and stemness properties	[159]
FEZF1-AS1	miR-30a	Nanog	Stemness, mammospheres, growth, proliferation, migration, and invasion	[160]
LINC00511	miR-185-3p	E2F1	Stemness, proliferation, mammospheres formation and tumor growth	[161]
LINC00617	Not apply	Sox2	Motility, invasion, EMT and metastasis	[162]

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
