# Peer review of "LncRNAs and microRNAs as Essential Regulators of Stemness in Breast Cancer Stem Cells"

_biomolecules, 2021, doi:10.3390/biom11030380_

Round 1

Reviewer 1 Report

This review article does not provide any novel findings to the current field of breast cancer. Secondly, the overall article is lengthy and lacks focused discussion on the role of lncRNA and miRNA on breast cancer stemness. 

Author Response

-Reply: Thank you very much for your comments. The goal of this review was not to show novel findings. In contrast, we provide here an actualization in an emerging topic in cancer summarizing the current status of CSCs in breast cancer. As you suggested we have better focused the topics of review and the discussion.

As the number of reported reviews about the roles of lncRNAs/miRNAs in breast cancer stem cells is very scarce (PubMed search:  only 4 review have been published in relation with lncRNAs/miRNAs in breast cancer stem cells in the last 10 years ! ), in this regard our review is opportune.

Reviewer 2 Report

The review written by Flores-Huerta and colleagues titled “LncRNAs and microRNAs as essential regulators of stemness in breast cancer stem cells” is a well-organized and thorough review of these two biologically active RNA molecule classes in the initiation and progression of breast cancer.  The introduction does an excellent job of describing what a true stem cell is – how it is defined and what its role in biology is – specifically mammary gland development.  The development of breast tumors and the signaling pathways known to regulate breast cell growth and proliferation are covered.  Finally, the roles of both microRNAs and lncRNAs are included in the development of breast cancer and are tied into the previously discussed signaling pathways. 

One major problem with the review – in the title as well as in the body of the review – is that the term “Cancer Stem Cell” is a loaded term with a very controversial history.  The wealth of data examining these cells especially in blood-based malignancies such as leukemias and myelomas have not clearly shown that “Cancer Stem Cells” actually exist.  This is even more true with studies examining solid tumors, such as breast cancer, which have not demonstrated their existence. A more acceptable term would be “Tumor Initiating Cell” which eliminates the problems associated with controversial “stem cell” description of this cell population.  Stem cells can become any type of cell in an organism as well described in the introduction of this review; however, Tumor Initiating Cells can only become tumors and lack the plasticity of true stem cells.  In addition, Tumor Initiating Cells are genomically unstable due to acquired mutations compared to true stem cells.  There is a large body of evidence in the literature that what have been described as “Cancer Stem Cells” are actually tumor cells that are dedifferentiating, but never return to the stem cell state limiting their potential to act as a true stem cell.  It would be a great service to the body of scientific literature to stop using “Cancer Stem Cells” due to the imprecise nature of the term and the confusion it creates.

A specific point related to this controversy is presented on page 7, line 310 - stem cells have not been shown to go through the process of EMT; however, the review states “…bCSC differentiated trough induction of EMT…”.  This is incorrect.

ADAR1 is an RNA editase that has been recently shown to play an important role in breast cancer, specifically the triple negative breast cancer subtype.  Given its enzymatic function to edit RNAs changing Adenosines to Inosines and the data demonstrating its RNA targets are often non-coding RNAs, discussion of ADAR1’s known roles in altering microRNAs and lncRNAs to alter their function in breast cancer initiation and progression should be included in this review.  In addition, the JAK/STAT signaling pathway (section 4.5) has been shown to regulate ADAR1 expression and function as part of the Type I Interferon Innate Immune Response.

There are several places throughout the review that the English grammar and word choice is extremely confusing.  This requires the reader to guess as to what the sentence or idea means which should never be done.  It would help to have a native English speaker edit the review with the authors to ensure the ideas presented are unambiguous.

Author Response

We are grateful and greatly appreciate the time of reviewers to evaluate our manuscript. In this revised version, we have answered point by point the comments and concerns which improve notably the quality of manuscript. All changes made were highlighted with yellow in the revised version of manuscript for easy identification.

Comments and Suggestions for Authors

The review written by Flores-Huerta and colleagues titled “LncRNAs and microRNAs as essential regulators of stemness in breast cancer stem cells” is a well-organized and thorough review of these two biologically active RNA molecule classes in the initiation and progression of breast cancer.  The introduction does an excellent job of describing what a true stem cell is – how it is defined and what its role in biology is – specifically mammary gland development.  The development of breast tumors and the signaling pathways known to regulate breast cell growth and proliferation are covered.  Finally, the roles of both microRNAs and lncRNAs are included in the development of breast cancer and are tied into the previously discussed signaling pathways.

  1. One major problem with the review – in the title as well as in the body of the review – is that the term “Cancer Stem Cell” is a loaded term with a very controversial history.  The wealth of data examining these cells especially in blood-based malignancies such as leukemias and myelomas have not clearly shown that “Cancer Stem Cells” actually exist.  This is even more true with studies examining solid tumors, such as breast cancer, which have not demonstrated their existence. A more acceptable term would be “Tumor Initiating Cell” which eliminates the problems associated with controversial “stem cell” description of this cell population.  Stem cells can become any type of cell in an organism as well described in the introduction of this review; however, Tumor Initiating Cells can only become tumors and lack the plasticity of true stem cells.  In addition, Tumor Initiating Cells are genomically unstable due to acquired mutations compared to true stem cells. There is a large body of evidence in the literature that what have been described as “Cancer Stem Cells” are actually tumor cells that are dedifferentiating, but never return to the stem cell state limiting their potential to act as a true stem cell.  It would be a great service to the body of scientific literature to stop using “Cancer Stem Cells” due to the imprecise nature of the term and the confusion it creates.

-Reply: Thank you very much for your observation and we appreciate the comments, as you mentioned it could be correct to name the Cancer Stem Cells-like (CSC) as “Tumor Initiating Cell”, however for our understanding both terms are synonyms. We know about the controversy of CSC mainly about its origin and tumor initiation. We have noted in the literature that there is a confusion in the terms. For us, and the majority of cancer researchers, Cancer Stem Cells-like refers to genetically unstably mature cells that acquired mutations and epigenetic alterations driving the carcinogenesis, and which are able to form tumors in animals and that, as results of the deregulated expression in the early stages of tumorigenesis, they expressed a set of genes related with stemness properties and abilities,. This notion not mean that CSC are truly primordial Stem Cells that evolved to a transformed and cancerous state (although this a current hypothesis, which until today have not been refuted). In contrast, not all the “Tumor Initiating Cells” show the expression of stemness genes. One hypothesis of the cancer stem cell-like origin postulates that cancer arises from a subpopulation of tumor initiating cells or CSC. It’s true that these concepts were initially described in hematological malignancies, and then confirmed similar mechanisms in solid tumors.

Currently, its widely accepted by the scientific community that CSC subpopulations are defined by the expression of Stem Cells markers (CD44, CD24, ALDH1, CD133, Nanog, Sox2, Oct-4, c-Myc etc) and showed phenotypic characteristics indicative of stemness properties such as self-renewal, spheroids formation, increased resistance to chemotherapy and formation of tumors in animal models; which reinforces the idea that CSC are somatic cells that acquire a mutant stem cell-like phenotype, but not a normal Stem Cell that acquire a tumoral behavior (Cantile et al. 2013; Charafe-Jauffret et al. 2010; Ghanei et al. 2020; Ginestier et al. 2007; Kim and Nam 2011; Kristiansen et al. 2003; Li et al. 2017; Ricardo et al. 2011)

In breast cancer, it has been demonstrated that breast cancer stem cells in tumors contain high levels of CD44 and low levels of CD24 membrane proteins. Although the CSC with an CD44+CD24-/low immunophenotype represent a minor subpopulation in the bulk tumor, these cells have the ability to produce efficiently solid tumors in mice, and to propagate in vitro with extensive cell proliferation and self-renewal abilities (Al-Hajj et al. 2003; Ponti et al. 2005). Hypotheses regarding the origins of CSCs include: (1) malignant transformation of normal stem cells; (2) mature somatic cancer cell de-differentiation with epithelial mesenchymal transition; and (3) induced pluripotent cancer cells. Surprisingly, the cancer stem cell hypothesis originated in the late nineteenth century and their existence was demonstrated a century later, demonstrates that the concept was possible. (Butti et al. 2019; Islam et al. 2015; Taurin and Alkhalifa 2020).

We agree with the reviewer comment that “there is a large body of evidence in the literature that “Cancer Stem Cells” are actually tumor cells that are dedifferentiating, but never return to the stem cell state limiting their potential to act as a true stem cell”. Thus, it’s true that CSC never will return to the multipotent state; however, these cells may keep a cancer stem cell-like phenotype supported by the aberrant expression of stemness genes.

We have rewritten the abstract section to clarify these concepts, and also shortened the information about the normal stem cells involved in the development of mammary gland to avoid confusion with the origin of CSC. Therefore, in this review, we will use the terms Cancer Stem Cells and Tumor Initiating Cells as synonyms as this the most widely accepted by cancer researchers.

Also, we have rewritten the abstract (lanes 18-34) and introduction (lanes 48-64) to clarify these points as follows: Currently, a hypothesis of the origin of cancer is based on the alterations occurring in primordial mammary stem cells producing small subpopulations of “cancer stem cells-like” (CSC) or “tumor initiating cells” as essential protagonists of the aberrant behavior driving tumor initiation, chemoresistance and recurrence is regaining strength. This hypothesis has been difficult to confirm, and an alternative hypothesis declare that CSC are originated from genetic and epigenetic alterations in somatic differentiated cells which drives the activation of a set of genes involved in pluripotency, stemness and development during the early stages to tumorigenesis. In addition, currently there is also controversy on the exact nomenclature for these cells with stemness properties as most of researches denominates these subpopulations as cancer stem cells-like or alternatively as tumor initiating cells (here in this review we will refers to both subpopulations types as synonymous). Thus, the origin of CSC is uncertain, however three possible hypotheses are viable: i) the malignant transformation of normal stem cells, ii) cancer cell de-differentiation with epithelial-mesenchymal transition (EMT) and; iii) induced pluripotent cancer cells (Islam et al. 2015). In the same way that stem cells exist in normal tissues, the CSC are also present in the tumorigenic tissues which possibly were originated by genetic and epigenetic mutations in the normal stem cells promoting specific failure in the processes that regulates cell differentiation (Lau et al. 2017). Independently of its origin, the CSC are defined as cells with the ability to preserve themselves as result of self-renewal, and with the capability to generate mature cells of determined tissue trough cell differentiation (Lau et al. 2017; Reya et al. 2001).

  1. A specific point related to this controversy is presented on page 7, line 310 - stem cells have not been shown to go through the process of EMT; however, the review states “…bCSC differentiated trough induction of EMT…”.  This is incorrect.

-Reply: Thank you very much, we appreciate the comment. We agree with the reviewer and eliminated this paragraph because is incorrect. We apologize for the confusion.

  1. ADAR1 is an RNA editase that has been recently shown to play an important role in breast cancer, specifically the triple negative breast cancer subtype.  Given its enzymatic function to edit RNAs changing Adenosines to Inosines and the data demonstrating its RNA targets are often non-coding RNAs, discussion of ADAR1’s known roles in altering microRNAs and lncRNAs to alter their function in breast cancer initiation and progression should be included in this review.  In addition, the JAK/STAT signaling pathway (section 4.5) has been shown to regulate ADAR1 expression and function as part of the Type I Interferon Innate Immune Response.

- Reply: Thank you very much for your valuable suggestion, we added the requested information about the functional roles of ADAR1 in the section 4.5, pag. 11, as follows:

“Finally, it was reported that high ADAR1 expression was associated with immune responses. Moreover, JAK/STAT signaling was differentially regulated in basal-like tumors with high ADAR1 expression (Fumagalli et al. 2015; Song et al. 2017). Adenosine deaminases acting on RNA (ADARs) catalyzing the edition of adenosine-to-inosine (A-to-I) in double stranded RNA, a common post-transcriptional modification in mammals. ADAR1 catalytically active enzyme is ubiquitously expressed in many tissues. Inosine is interpreted as guanosine during base-pairing, which can lead to codon changes with the consequence of altered protein function, affect targeting and maturation of microRNAs. thus, it has been shown in several studies that the transcriptomic as well as the proteomic diversity introduced by A-to-I editing is exploited by tumor cells to promote cancer progression; for example, more editing events regulated by ADAR1 have been associated with cancer development, primarily due to more abundant expression of ADAR1in BC including triple negative BC, that is required for its survival. Knockdown of ADAR1 attenuates proliferation and tumorigenesis, leads to robust translational repression (Kung et al. 2020; Kung et al. 2021; Xu and Öhman 2018). Recently, it has been evaluated the role ADAR1 and lncRNAs interplay in cancer, suggesting that ADAR1 could alter the expression levels of lncRNAs and explore how those changes are related to BC biology. Basically, the study show that LINC00944 is responsive to ADAR1 up- and downregulation in breast cancer cells, LINC00944 expression has a strong relationship with immune signaling and LINC00944 expression was correlated to the age at diagnosis, tumor size, and estrogen and progesterone receptor expression. Finally, low expression of LINC00944 is correlated to poor prognosis in BC patients (de Santiago et al. 2020). Additionally, ADAR enzymes modify a large proportion of cellular RNAs, contributing to transcriptome diversity and cancer evolution. It has demonstrated that exist an increased editing at 3'UTR in breast cancer cells compared to immortalized non-malignant cell lines, the results suggest a significant association between the mRNA editing in genes related to cancer-relevant pathways, suggesting an significant role of ADAR1 in BC (Sagredo et al. 2018)”.

  1. There are several places throughout the review that the English grammar and word choice is extremely confusing.  This requires the reader to guess as to what the sentence or idea means which should never be done.  It would help to have a native English speaker edit the review with the authors to ensure the ideas presented are unambiguous.

-Reply: Thank you very much for your valuable comments, we have corrected the manuscript for style correction by a native English speaker using the MDPI language services (# ID 26841).

References

Al-Hajj, M., Wicha, M.S., Benito-Hernandez, A., Morrison, S.J., Clarke, M.F. (2003). Prospective identification of tumorigenic breast cancer cells. Proceedings of the National Academy of Sciences of the United States of America, 100, 3983-3988.

Butti, R., Gunasekaran, V.P., Kumar, T.V.S., Banerjee, P., Kundu, G.C. (2019). Breast cancer stem cells: Biology and therapeutic implications. The International Journal of Biochemistry & Cell Biology, 107, 38-52.

Cantile, M., Collina, F., D'Aiuto, M., Rinaldo, M., Pirozzi, G., Borsellino, C., Franco, R., Botti, G., Di Bonito, M. (2013). Nuclear Localization of Cancer Stem Cell Marker CD133 in Triple-Negative Breast Cancer: A Case Report. Tumori Journal, 99, e245-e250.

Charafe-Jauffret, E., Ginestier, C., Iovino, F., Tarpin, C., Diebel, M., Esterni, B., Houvenaeghel, G., Extra, J.M., Bertucci, F., Jacquemier, J., Xerri, L., Dontu, G., Stassi, G., Xiao, Y., Barsky, S.H., Birnbaum, D., Viens, P., Wicha, M.S. (2010). Aldehyde dehydrogenase 1-positive cancer stem cells mediate metastasis and poor clinical outcome in inflammatory breast cancer. Clin Cancer Res, 16, 45-55.

de Santiago, P.R., Blanco, A., Morales, F., Marcelain, K., Harismendy, O., Sjöberg Herrera, M., Armisén, R. (2020). Immune-related IncRNA LINC00944 responds to variations in ADAR1 levels and it is associated with breast cancer prognosis. Life Sci, 268, 118956.

Fumagalli, D., Gacquer, D., Rothé, F., Lefort, A., Libert, F., Brown, D., Kheddoumi, N., Shlien, A., Konopka, T., Salgado, R., Larsimont, D., Polyak, K., Willard-Gallo, K., Desmedt, C., Piccart, M., Abramowicz, M., Campbell, P.J., Sotiriou, C., Detours, V. (2015). Principles Governing A-to-I RNA Editing in the Breast Cancer Transcriptome. Cell Rep, 13, 277-289.

Ghanei, Z., Jamshidizad, A., Joupari, M.D., Shamsara, M. (2020). Isolation and characterization of breast cancer stem cell-like phenotype by Oct4 promoter-mediated activity. Journal of Cellular Physiology, 235, 7840-7848.

Ginestier, C., Hur, M.H., Charafe-Jauffret, E., Monville, F., Dutcher, J., Brown, M., Jacquemier, J., Viens, P., Kleer, C.G., Liu, S., Schott, A., Hayes, D., Birnbaum, D., Wicha, M.S., Dontu, G. (2007). ALDH1 is a marker of normal and malignant human mammary stem cells and a predictor of poor clinical outcome. Cell stem cell, 1, 555-567.

Islam, F., Qiao, B., Smith, R.A., Gopalan, V., Lam, A.K. (2015). Cancer stem cell: fundamental experimental pathological concepts and updates. Exp Mol Pathol, 98, 184-191.

Kim, R.-J., Nam, J.-S. (2011). OCT4 Expression Enhances Features of Cancer Stem Cells in a Mouse Model of Breast Cancer. Laboratory animal research, 27, 147-152.

Kristiansen, G., Winzer, K.-J., Mayordomo, E., Bellach, J., Schlüns, K., Denkert, C., Dahl, E., Pilarsky, C., Altevogt, P., Guski, H., Dietel, M. (2003). CD24 Expression Is a New Prognostic Marker in Breast Cancer. Clinical cancer research, 9, 4906-4913.

Kung, C.-P., Cottrell, K.A., Ryu, S., Bramel, E.R., Kladney, R.D., Bross, E.A., Maggi, L., Weber, J.D. (2020). ADAR1 editing dependency in triple-negative breast cancer. bioRxiv, 2020.2001.2031.928911.

Kung, C.P., Cottrell, K.A., Ryu, S., Bramel, E.R., Kladney, R.D., Bao, E.A., Freeman, E.C., Sabloak, T., Maggi, L., Jr., Weber, J.D. (2021). Evaluating the therapeutic potential of ADAR1 inhibition for triple-negative breast cancer. Oncogene, 40, 189-202.

Lau, E.Y.-T., Ho, N.P.-Y., Lee, T.K.-W. (2017). Cancer Stem Cells and Their Microenvironment: Biology and Therapeutic Implications. Stem cells international, 2017, 3714190-3714190.

Li, W., Ma, H., Zhang, J., Zhu, L., Wang, C., Yang, Y. (2017). Unraveling the roles of CD44/CD24 and ALDH1 as cancer stem cell markers in tumorigenesis and metastasis. Scientific Reports, 7, 13856.

Ponti, D., Costa, A., Zaffaroni, N., Pratesi, G., Petrangolini, G., Coradini, D., Pilotti, S., Pierotti, M.A., Daidone, M.G. (2005). Isolation and in vitro propagation of tumorigenic breast cancer cells with stem/progenitor cell properties. Cancer Res, 65, 5506-5511.

Reya, T., Morrison, S.J., Clarke, M.F., Weissman, I.L. (2001). Stem cells, cancer, and cancer stem cells. Nature, 414, 105-111.

Ricardo, S., Vieira, A.F., Gerhard, R., Leitão, D., Pinto, R., Cameselle-Teijeiro, J.F., Milanezi, F., Schmitt, F., Paredes, J. (2011). Breast cancer stem cell markers CD44, CD24 and ALDH1: expression distribution within intrinsic molecular subtype. Journal of clinical pathology, 64, 937-946.

Sagredo, E.A., Blanco, A., Sagredo, A.I., Pérez, P., Sepúlveda-Hermosilla, G., Morales, F., Müller, B., Verdugo, R., Marcelain, K., Harismendy, O., Armisén, R. (2018). ADAR1-mediated RNA-editing of 3'UTRs in breast cancer. Biol Res, 51, 018-0185.

Song, I.H., Kim, Y.-A., Heo, S.-H., Park, I.A., Lee, M., Bang, W.S., Park, H.S., Gong, G., Lee, H.J. (2017). ADAR1 expression is associated with tumour-infiltrating lymphocytes in triple-negative breast cancer. Tumor Biology, 39, 1010428317734816.

Taurin, S., Alkhalifa, H. (2020). Breast cancers, mammary stem cells, and cancer stem cells, characteristics, and hypotheses. Neoplasia (New York, N.Y.), 22, 663-678.

Xu, L.D., Öhman, M. (2018). ADAR1 Editing and its Role in Cancer. Genes, 10.

Reviewer 3 Report

The topic discussed in the article is very current, important and clearly interesting. I appreciate the attempt to summarize the issue in a very detail and comprehensive way.

However, the extension of the individual paragraphs is rather detrimental to the quality of a whole work. For example, the paragraph: Biology of breast cancer stem cells and their main molecular characteristics - does not describe issues that would be directly related to the following parts of the article and is therefore only an extension of the introduction focusing on well-known and many times described aspects of histology, physiology and molecular biology. Paragraph: Origin of breast cancer stem cell - does not describe the process of cancer stem cell formation, nor EMT. In the row 228 there are mentioned three currently accepted hypotheses about the origin of breast cancer stem cell, but in the following text only two are discussed; row 254 reads: The function of CD44 CD24… - however, the function is not described further, some of the factors involved in cell transformation (Oct-4, ALDH1) are described identically in several previous paragraphs. The sections devoted to individual signalling pathways do not describe the mechanism of the pathways precisely and in relation to malignant transformation only very superficially. Regarding the role of non-coding RNAs in the modulation of signalling, the description of individual pathways is insufficient - in some cases the description of key pathways and molecules is completely missing (PTEN / AKT; p53). In contrast to the description of signal transduction pathways, the mechanisms of action of miRNAs and lncRNAs are completely absent. Their effect on cell transformation is then difficult to explain, and the relevant paragraphs contain only a list of many known miRNAs and lncRNAs with a superficial description of their targets. The clinical aspect of the role of non-coding RNAs in cell transformation is not discussed at all in the article. The used figures are quite uninformative and do not contribute to the understanding of the described issues. The quality of their processing is low (the size of the image does not correspond to the amount of information that the reader can obtain from the image, the font is small, blurred and barely readable). In addition, the article contains a number of inaccuracies: line 44 - breast cancer is not the second most common neoplasia; the stem cell / cancer stem cell / breast cancer stem cell are replaced several times in the article; row 261 - transcription factor Oct-4 itself is not a major regulator. Conclusions are strongly disproportional to other parts of the article and do not point out the most important and interesting aspects of discussed problematics. I consider the low level of English to be the biggest shortcoming of the article - the meaning of a number of sentences is very difficult to understand or bad.

With all respect to the effort of authors I consider the above mentioned shortcomings of the article to be very significant and the article in its current form cannot be published. I therefore suggest reconsidering the acceptance of the article for publication after extensive modifications.

Author Response

We are grateful and greatly appreciate the time of reviewers to evaluate our manuscript. In this revised version, we have answered point by point the comments and concerns which improve notably the quality of manuscript. All changes made were highlighted with yellow in the revised version of manuscript for easy identification.

Thank you very much for your valuable corrections and comments, we have analyzed and reviewed point by point your corrections, undoubtedly our manuscript will improve notably.

Comments and Suggestions for Authors

  1. The topic discussed in the article is very current, important and clearly interesting. I appreciate the attempt to summarize the issue in a very detail and comprehensive way. However, the extension of the individual paragraphs is rather detrimental to the quality of a whole work. For example, the paragraph: Biology of breast cancer stem cells and their main molecular characteristics - does not describe issues that would be directly related to the following parts of the article and is therefore only an extension of the introduction focusing on well-known and many times described aspects of histology, physiology and molecular biology.

-Reply: Thank you very much for your observation and we appreciate the comments, as you mentioned is correct, we changed the title of this section because we considered that it was not adequate. 2. The mammary gland: main biological and molecular characteristics.

In addition, we have shortened this section as the goal was only provide a general view of the genes and mechanisms governing the mammary gland normal development, as many of these genes are reactivated in the somatic mammary cells during the tumorigenesis originating the CSCs o Tumor Initiating Cells.

  1. Paragraph: Origin of breast cancer stem cell - does not describe the process of cancer stem cell formation, nor EMT.

-Reply: Thank you very much for your observations we appreciate the comments. We changed the title of this section because we considered that it was more adequate. 3. Breast cancer stem cells origin and their main cell markers. To clarify the origin of CSC we added information about the current 3 hypotheses: (1) malignant transformation of normal stem cells; (2) mature cancer cell de-differentiation with epithelial-mesenchymal transition (EMT) and (3) induced pluripotent cancer cells. Also, we have clarify the EMT processes associated to stemness in CSC as follows:

“As known, the cancer is the result of several genetic and epigenetic alterations that cause production of growth-related genes and to arise from a single cell in specific tissues, for example the breast. Altering the mechanisms that normally control a stable physiological cell promoting the genomic instability state (Islam et al. 2015). The precise bCSC origin is an ambiguous, and it has been controversially debated for long time. Exist several features similar between cancer cells and bCSC. Both can self-renew and share signaling pathways associated with the cell replication, recurrence and maintenance (Taurin and Alkhalifa 2020). In general, there are three main hypotheses for the acquisition of the properties of a SC for cancer cells. These include the hypotheses of (1) malignant transformation of normal stem cells; (2) mature cancer cell de-differentiation with epithelial-mesenchymal transition (EMT) and (3) induced pluripotent cancer cells (Islam et al. 2015; Sin and Lim 2017). Particularly, for BC two nonexclusive more accepted models have been proposed to explain clonal populations in the tumor, imply the accumulation of independent mutational events and likely that both models coexist. The first model involves the stochastic appearance of mutations and clonal selection that grant the cells stem-like properties and ability to differentiate and self-renew. In the second model, the MaSC and progenitor attributes are central to the heterogeneity of the breast cancer cell populations. The accumulation of these genetic and epigenetic alterations results in the development of at least one cell with CSC traits that can produce more CSCs and more differentiated offspring. In the past, the CSC model was a static one. However, in recent times, it has been revised to a dynamic one, where CSCs were believe convert into more transient cell types (Butti et al. 2019; Taurin and Alkhalifa 2020). It is important to mention the cancer cells heterogeneity, not all cancer cells are stem cells or exhibit properties similar to stem cells. The cell diversity and heterogeneity is product of mutagenesis present in cancer cells and result in incomplete or aberrant hierarchical cellular differentiation. Therefore, according with the clonal evolution/stochastic all the tumoral cells have a similar tumorigenic potential and tumor heterogeneity arises as a result of the generation of intra-tumoral clones through the sequential mutations. This model presumes that bCSC can be generated from differentiated mammary cells by virtue of mutations that occur in course of the disease. Exposure to detrimental environmental factors such as radiation and chemotherapies induce genetic alterations in non-malignant somatic cells that prime the de novo generation of bCSC by the de-differentiation process, and even, the microenvironmental signals induce the malignant transformation of differentiated cells into bCSC. Hierarchical or CSC model postulates that only a small proportion of tumor cells reside in the tumor has a tumor-propagating potential. These cells exhibit self-renewal properties and are capable of reiterating tumor hierarchy (Butti et al. 2019; Sin and Lim 2017; Taurin and Alkhalifa 2020). In addition, the current gold standard method to identify CSC is via serial transplantation of cancer cells in animal models. In BC this it has corroborate, the isolation of bCSC and the subsequent transplantation. In this regard, bCSC with an CD44+/CD24-/low immunophenotype produce efficiently solid tumors in mice, CD44+/CD24 signature is a molecular determinant of bCSC (Al-Hajj et al. 2003; Ponti et al. 2005). Studies have suggested a close association between CSCs and the acquisition of an EMT state (Mani et al. 2008). Interestingly, bCSCs exist in distinct mesenchymal-like EMT and mesenchymal-epithelial transition (epithelial-like, MET) states. Mesenchymal-like bCSC characterized as CD24/CD44+ are primarily quiescent and localized at the tumor invasive front, whereas epithelial-like bCSC express ALDH proliferative, and are located more centrally (Liu et al. 2014). However, further studies are needed to more fully define the relationship among EMT, MET, and bCSC”.

  1. In the row 228 there are mentioned three currently accepted hypotheses about the origin of breast cancer stem cell, but in the following text only two are discussed.

-Reply: Thank you very much for your observation, we apologize for the syntaxes mistake. The 3 hypothesis were already discussed but we omitted the word “third”, as the occurrence of mutations in MaSC or progenitors that are central to the heterogeneity of the BC cell populations (Butti et al. 2019; Islam et al. 2015; Taurin and Alkhalifa 2020). We have corrected the text.

  1. Row 254 reads: The function of CD44 CD24… - however, the function is not described further,

-Reply: Thank you very much for your observation and we appreciate the comments, as you requested, we have added the summarized information about the roles of CD44 and CD 24 proteins in the corresponding section.

CD44 is a glycoprotein that act as an adhesion molecule mainly found in primary breast carcinomas that is aberrantly expressed depending to levels of cell differentiation. The BC with a basal phenotype expressed a high percentage of bCSC with CD44+/CD24-/low immunophenotype because probably originated from the most primitive mammary stem cells, contrarily to the luminal phenotype that shown a cell population enrichment with CD44-/CD24+ (Honeth et al. 2008; Park et al. 2010; Ricardo et al. 2011).

CD24 is a small, heavily glycosylated mucin-like glycosylphosphatidylinositol-linked cell surface protein that is expressed in a wide variety of human malignancies, including the BC. Is denomined as prognosis marker, due CD24 expression might enhance the metastatic potential of tumor cells, because CD24 has been identified as an alternative ligand of P-selectin (Kristiansen et al. 2003).

  1. Some of the factors involved in cell transformation (Oct-4, ALDH1) are described identically in several previous paragraphs.

-Reply: Thank you very much for your observation and we appreciate the comments. We have checked the complete manuscript and we did not found description of Oct-4 and ALDH1 as equally.

  1. The sections devoted to individual signaling pathways do not describe the mechanism of the pathways precisely and in relation to malignant transformation only very superficially.

-Reply: Thank you very much for your observation, we appreciate the comments and we are sorry that you have not transmitted what we wanted to capture and describe. We edited the information to remark the pathways in the malignant transformation. We wanted to adequately synthesize all the existing information highlight that in bCSC, the deregulation of these pathways is one the main reasons for the appearance or exacerbation of the main hallmarks such as maintenance, self-renewal, tumor resistance, recurrence, and metastasis of bCSC.

  1. Regarding the role of non-coding RNAs in the modulation of signalling, the description of individual pathways is insufficient - in some cases the description of key pathways and molecules is completely missing (PTEN/AKT; p53).

-Reply: Thank you very much for your observation and we appreciate the comments. Regarding to the role of non-coding RNAs in the signalling, we wanted highlight its importance and for this purpose edited this. The description of these molecules is include in the section corresponding. For example, phosphatase and tensin homolog (PTEN), AKT, protein kinase B (PKB) and this regulation also is controlled by p53 gene, which is a tumor suppressor that regulates the cellular division, and also called the "guardian of the genome".

  1. In contrast to the description of signal transduction pathways, the mechanisms of action of miRNAs and lncRNAs are completely absent. Their effect on cell transformation is then difficult to explain, and the relevant paragraphs contain only a list of many known miRNAs and lncRNAs with a superficial description of their targets. The clinical aspect of the role of non-coding RNAs in cell transformation is not discussed at all in the article.

-Reply: Thank you very much for your observation and we appreciate the comments. We add this information of the mechanisms of action of miRNAs and lncRNAs in a summarized way in the correspond section.

-The function is based in binding to complementary target sequences in messenger RNA (mRNA) and interfering with the translational machinery, thereby preventing or altering the production of the protein product (repressing translation). miRNA binding to its target mRNA also triggered the recruitment and association of mRNA decay factors, leading to mRNA destabilization and degradation (Bhaskaran and Mohan 2014).

-LncRNAs constitute another important class of non-coding RNAs with lengths exceeding 200 nucleotides; as a particular characteristic, they are not translated into protein. Exerting the regulation of diverse genes at the epigenetic, transcriptional, and post-transcriptional level through the interaction with miRNAs, mRNAs, proteins, and genomic DNA. Thus, lncRNAs are actively involved in chromatin rearrangement, histone modification, regulation of transcription, regulation of basal transcription machinery, post-transcriptional regulation splicing translation, as well as regulation of gene expression (Soudyab et al. 2016).

-The information of the several reports that support the fundamental role of miRNAs and lncRNAs in the bCSC hallmarks is described in their respective section. Due to the vast amount of existing information, we wanted to include the most relevant articles in a summarized way.The table you mention is a summary complement of what is described in the article with its respective citations, in case the reader is interested in a particular article, he can consult it.

  1. The used figures are quite uninformative and do not contribute to the understanding of the described issues. The quality of their processing is low (the size of the image does not correspond to the amount of information that the reader can obtain from the image, the font is small, blurred and barely readable).

-Reply: Thank you very much for your observation and we appreciate the comments, as you mentioned the information about the bCSC is considerable and it is not appropriate to fill with figures of each signaling pathway, likewise it is inappropriate to put all the pathways in a single figure with all the molecules that participate, it would be very loaded. So we wanted to synthesize the information to only those routes in which both the pathway, the miRNA, lncRNA and the target are described and the mechanisms in which they impact. A small synthesis of the complete information of each article is described in each section on miRNAs and lncRNA, the two images are complemented with information and citations. It is impossible to describe the entire article since we wanted to include the vast majority of bCSC reports. Finally, we adjust to the requirements of the journal which mentions according with the instructions for authors, resolution (minimum 1000 pixels width/height, or a resolution of 300 dpi or higher) and smaller fonts may be used, but no less than 8 pt in size. Now in the corrected version, we edited the figure at 18 pt in size of font and 300 dpi.

  1. In addition, the article contains a number of inaccuracies: line 44 - breast cancer is not the second most common neoplasia.

-Reply: Several recent publications report to the breast cancer between the first and second most common. Thus, according with globocan 2018-2019 estimates of cancer incidence and mortality in both sexes combined, lung cancer is the most commonly diagnosed cancer with 2,093,876 new cases (11.6% of the total cases) and the leading cause of cancer death 1,761,007 (18.4% of the total cancer deaths), closely followed by female breast cancer with 2,088,849 (11.6%), and 626,679 (6.6 of the total cancer deaths). Among females, breast cancer is the most commonly diagnosed cancer and the leading cause of cancer death, with first incidence only in females is of 24.2% and mortality of 15.0% (Bray et al. 2018; Sun et al. 2017) https://www.bcrf.org/breast-cancer-statistics-and-resources

  1. the stem cell / cancer stem cell / breast cancer stem cell are replaced several times in the article; row 261 - transcription factor Oct-4 itself is not a major regulator. Conclusions are strongly disproportional to other parts of the article and do not point out the most important and interesting aspects of discussed problematic. I consider the low level of English to be the biggest shortcoming of the article - the meaning of a number of sentences is very difficult to understand or bad.

-Reply: Thank you very much for your valuable comments. We review these terms and are adequate. We defined stem cells (SC) can refer to stem cell in general, cancer stem cell (CSC) can refer to any type of cancer and particularly (bCSC) to breast cancer stem cell. Because the topic of the article is related to the bCSC, through the article these terms are repeated.

-Row 261: we now described as follow: “Oct-4 is a master regulator fundamental in the self-renewal and pluripotency, associated with maintenance and expansion of CSC” (Ghanei et al. 2020; Kim and Nam 2011; Ponti et al. 2005). However, certainly also exist another transcription factors characteristic of stemness and pluripotency, as well as, Nanog, Sox, Myc, that were described at the end of section 2.

-Regarding to conclusion, we consider that remark the importance of bCSC study and the participation of miRNA and lncRNA, these molecules are promising for the control and treatment of the breast cancer and others neoplasia.

We modified the conclusion as follows:

Several studies support that bCSC are responsible for the propagation, chemo-resistance, and recurrence of BC, thus, supporting its protagonist in the cancer pathogenesis and provided an attractive area of research. In an early future enhanced outcome in BC patients will be possible, the personalized medicine supports the importance of the understanding of effectively and specifically by targeting bCSC.

It has been demonstrated that miRNAs and lncRNAs exert influence in the cancer development, such as tumorigenesis, apoptosis, metastasis, chemoresistance, radioresistance and angiogenesis; and are critical regulatory elements in the epigenetic mechanism of regulates of bCSC biology ad plasticity. Even, it has been suggested as biomarkers of cancer diagnosis and prognosis elements in patients with BC. The clinical trials support that the miRNAs expression of been suggested as biomarkers for early detection and therapeutic response monitoring in BC patients. LncRNAs and miRNAs represent attractive molecular targets and molecular tools that might be used for bCSC directed therapy. Further studies are necessary for to increase the understanding of the complex networks involved in controlling stem cells modulated by miRNAs and lncRNAs. These investigations promising allow discover the molecular mechanisms corrupted by which differentiated cancer cells and bCSC promote the BC apparition. Although much remains to be under-stood, future studies around the mechanisms, expression, targets, etc., that miRNAs and lncRNAs modulate would us provide information on the mechanism that these molecules controlling in the bCSCs. Even, the development of adjuvant therapy using the standard treatment, lncRNA/miRNAs, and natural compounds, can be a promising strategy against BC.

-Finally, we have sent the manuscript for style correction by a native English speaker.

References

Al-Hajj, M., Wicha, M.S., Benito-Hernandez, A., Morrison, S.J., Clarke, M.F. (2003). Prospective identification of tumorigenic breast cancer cells. Proceedings of the National Academy of Sciences of the United States of America, 100, 3983-3988.

Bhaskaran, M., Mohan, M. (2014). MicroRNAs: history, biogenesis, and their evolving role in animal development and disease. Veterinary pathology, 51, 759-774.

Bray, F., Ferlay, J., Soerjomataram, I., Siegel, R.L., Torre, L.A., Jemal, A. (2018). Global cancer statistics 2018: GLOBOCAN estimates of incidence and mortality worldwide for 36 cancers in 185 countries. CA: A Cancer Journal for Clinicians, 68, 394-424.

Butti, R., Gunasekaran, V.P., Kumar, T.V.S., Banerjee, P., Kundu, G.C. (2019). Breast cancer stem cells: Biology and therapeutic implications. The International Journal of Biochemistry & Cell Biology, 107, 38-52.

Ghanei, Z., Jamshidizad, A., Joupari, M.D., Shamsara, M. (2020). Isolation and characterization of breast cancer stem cell-like phenotype by Oct4 promoter-mediated activity. Journal of Cellular Physiology, 235, 7840-7848.

Honeth, G., Bendahl, P.O., Ringnér, M., Saal, L.H., Gruvberger-Saal, S.K., Lövgren, K., Grabau, D., Fernö, M., Borg, A., Hegardt, C. (2008). The CD44+/CD24- phenotype is enriched in basal-like breast tumors. Breast Cancer Res, 10, 17.

Islam, F., Qiao, B., Smith, R.A., Gopalan, V., Lam, A.K. (2015). Cancer stem cell: fundamental experimental pathological concepts and updates. Exp Mol Pathol, 98, 184-191.

Kim, R.-J., Nam, J.-S. (2011). OCT4 Expression Enhances Features of Cancer Stem Cells in a Mouse Model of Breast Cancer. Laboratory animal research, 27, 147-152.

Kristiansen, G., Winzer, K.-J., Mayordomo, E., Bellach, J., Schlüns, K., Denkert, C., Dahl, E., Pilarsky, C., Altevogt, P., Guski, H., Dietel, M. (2003). CD24 Expression Is a New Prognostic Marker in Breast Cancer. Clinical cancer research, 9, 4906-4913.

Liu, S., Cong, Y., Wang, D., Sun, Y., Deng, L., Liu, Y., Martin-Trevino, R., Shang, L., McDermott, Sean P., Landis, Melissa D., Hong, S., Adams, A., D’Angelo, R., Ginestier, C., Charafe-Jauffret, E., Clouthier, Shawn G., Birnbaum, D., Wong, Stephen T., Zhan, M., Chang, Jenny C., Wicha, Max S. (2014). Breast Cancer Stem Cells Transition between Epithelial and Mesenchymal States Reflective of their Normal Counterparts. Stem Cell Reports, 2, 78-91.

Mani, S.A., Guo, W., Liao, M.-J., Eaton, E.N., Ayyanan, A., Zhou, A.Y., Brooks, M., Reinhard, F., Zhang, C.C., Shipitsin, M., Campbell, L.L., Polyak, K., Brisken, C., Yang, J., Weinberg, R.A. (2008). The epithelial-mesenchymal transition generates cells with properties of stem cells. Cell, 133, 704-715.

Park, S.Y., Lee, H.E., Li, H., Shipitsin, M., Gelman, R., Polyak, K. (2010). Heterogeneity for Stem Cell–Related Markers According to Tumor Subtype and Histologic Stage in Breast Cancer. Clinical cancer research, 16, 876-887.

Ponti, D., Costa, A., Zaffaroni, N., Pratesi, G., Petrangolini, G., Coradini, D., Pilotti, S., Pierotti, M.A., Daidone, M.G. (2005). Isolation and in vitro propagation of tumorigenic breast cancer cells with stem/progenitor cell properties. Cancer Res, 65, 5506-5511.

Ricardo, S., Vieira, A.F., Gerhard, R., Leitão, D., Pinto, R., Cameselle-Teijeiro, J.F., Milanezi, F., Schmitt, F., Paredes, J. (2011). Breast cancer stem cell markers CD44, CD24 and ALDH1: expression distribution within intrinsic molecular subtype. Journal of clinical pathology, 64, 937-946.

Sin, W.C., Lim, C.L. (2017). Breast cancer stem cells-from origins to targeted therapy. Stem cell investigation, 4, 96-96.

Soudyab, M., Iranpour, M., Ghafouri-Fard, S. (2016). The Role of Long Non-Coding RNAs in Breast Cancer. Arch Iran Med, 19, 508-517.

Sun, Y.-S., Zhao, Z., Yang, Z.-N., Xu, F., Lu, H.-J., Zhu, Z.-Y., Shi, W., Jiang, J., Yao, P.-P., Zhu, H.-P. (2017). Risk Factors and Preventions of Breast Cancer. International journal of biological sciences, 13, 1387-1397.

Taurin, S., Alkhalifa, H. (2020). Breast cancers, mammary stem cells, and cancer stem cells, characteristics, and hypotheses. Neoplasia (New York, N.Y.), 22, 663-678.

Round 2

Reviewer 2 Report

The authors have satisfactorily answered all of the critiques and made the appropriate changes to the manuscript.

Reviewer 3 Report

I would like to thank the authors for clarifications and all the modifications they have done! I consider the current version of the article to be very interesting and readable. It nicely summarizes all the important aspects in great details. I haven’t found any significant shortcomings – just want the authors to consider omitting of the statement in row 182 – the MCF7 cells, though widely used as a breast cancer model cell line, are not the ideal model as it exists in many cellular modification and the conclusions drawn through their application are debatable in general. I feel the general statement as it is in the following row – 185 to be good enough. Please check the text for correct styles (in vitro/in vitro) too. Regarding of above mentioned, I recommend to accept the current version of the article after minor modifications.